# PRODEM: An annual series of summer DEMs (2019 through 2022) of the marginal areas of the Greenland Ice Sheet

Mai Winstrup[1], Heidi Ranndal[1], Signe Hillerup Larsen[2], Sebastian B. Simonsen[1], Kenneth D. Mankoff[2,3,4], Robert S. Fausto[2], Louise Sandberg Sørensen[1]

[1]DTU Space, Technical University of Denmark (DTU), Kgs. Lyngby, 2800, Denmark
[2]Geological Survey of Denmark and Greenland (GEUS), Copenhagen, 2100, Denmark
[3] Autonomic Integra LLC, New York, NY, 10025, USA
[4] NASA Goddard Institute for Space Studies, New York, NY, 10025, USA

*Correspondence to*: Mai Winstrup (maiwin@dtu.dk)

**Abstract.** Surface topography across the marginal zone of the Greenland Ice Sheet is constantly evolving in response to changing weather, season, climate and ice dynamics. Yet current Digital Elevation Models (DEMs) for the ice sheet are usually based on data from a multi-year period, thus obscuring these changes over time. Here we present four 500-meter resolution summer DEMs (PRODEMs) of the Greenland Ice Sheet marginal zone for 2019 through 2022. The PRODEMs cover the marginal zone from the ice edge to 50 km inland, hence capturing all Greenland outlet glaciers. Each PRODEM is based on data fusion of CryoSat-2 radar altimetry and ICESat-2 laser altimetry using regionally-varying Kriging of elevation anomalies relative to ArcticDEM. The PRODEMs are validated using Leave-One-Out Cross-Validation, and PRODEM19 is further validated against an external data set, showcasing their ability to correctly represent surface elevations within the associated spatially varying prediction uncertainties. We observe a general lowering of surface elevations during the four-year PRODEM period, but the spatial pattern of change is highly complex and with annual changes superimposed. The PRODEMs enable detailed studies of the marginal ice sheet elevation changes. With their high spatio-temporal resolution, the PRODEMs will be of value to a wide range of researchers and users studying ice sheet dynamics and monitoring how the ice sheet responds to changing environmental conditions. PRODEMs from summer 2019 through 2022 are available at https://doi.org/10.22008/FK2/52WWHG (Winstrup, 2023), and we plan to annually update the product henceforth.

## 1 Introduction

During the past decades, the Greenland Ice Sheet has lost mass at an increasing rate (Otosaka et al., 2023; Shepherd et al., 2020; Simonsen et al., 2021) contributing to globally rising sea levels (Moon et al., 2018; WCRP Global Sea Level Budget Group, 2018). The ice sheet marginal regions are particularly vulnerable to shifts in temperature and precipitation patterns: While the central part of the Greenland Ice Sheet has maintained relatively constant ice thicknesses during recent decades, the ice sheet margins have progressively thinned (Krabill et al., 2004; Sørensen et al., 2018). Coupled ice-sheet–climate models suggest that future ice loss from the Greenland Ice Sheet will primarily be driven by changes in surface mass balance, particularly increased ablation at the ice sheet margin due to atmospheric warming (Quiquet and Dumas, 2021).

The ice sheet marginal areas are characterized by an interplay of many complex processes, with changes in ice sheet topography impacting many of these. Over longer timescales, ice thinning will initiate a positive feedback loop affecting the surface mass balance: As elevations decrease, the surface is exposed to higher temperatures, leading to enhanced surface melt and further reductions in elevation (Aschwanden et al., 2019). Eventually, this will increase the extent of the ice sheet ablation zone, amplifying mass loss (Noël et al., 2019). The ice sheet geometry also affects local and regional weather patterns (Ettema et al., 2009). With orographic precipitation being the main driver of Greenland precipitation in coastal areas (Lenaerts et al., 2019), a changing topography may have a large impact on precipitation patterns, also influencing the surface mass balance. The dynamic response of the ice sheet to a warmer climate will also be affected. Extensive surface melting in the ice sheet margin

creates an active hydrological network at and below the ice sheet surface with water routing depending on surface topography and ice thickness, respectively, having impacts on ice flow over an extended area (Andrews et al., 2018; Maier et al., 2023). Changes in the marginal ice sheet geometry will also directly impact the ice dynamics, contributing to accelerated ice flow towards the margins, hastening ice loss (Wang et al., 2012). Comprehensive monitoring of surface elevation for the Greenland Ice Sheet marginal zone is essential for enhancing our understanding of all these dynamic processes. The development of accurate DEMs for the Greenland Ice Sheet, not least its periphery, will advance our understanding of ice sheet dynamics and its response to ongoing environmental changes, and consequent impact on global sea levels.

Significant efforts have been directed towards enhancing the accuracy and coverage of DEMs for the Greenland Ice Sheet. Due to its vast scale and remote location, two main techniques to produce large-scale DEMs are based on respectively satellite altimetry and photogrammetry of satellite imagery stereo pairs, or some combination thereof, and possibly appended with airborne data from the marginal areas.

The first DEMs covering the entire Greenland Ice Sheet were based on ERS-1 and Geosat satellite radar altimetry along with a variety of airborne radar and laser altimetry (Ekholm, 1996), and later refined by including e.g., additional aerial photogrammetry for improved coverage of the marginal areas (Bamber et al., 2001a). Several corrections must be applied to the data when constructing DEMs based on radar altimetry. The initial return signal (found by 'retracking' the radar waveforms) will arrive from the Point of Closest Approach (POCA), and the radar will therefore tend to measure local topographic highs, causing a slope-induced bias in measured mean elevations (Bamber, 1994; Hurkmans et al., 2012; Levinsen et al., 2016). Further, the radar footprint is too large to produce reliable data in areas of highly varying topography (Brenner et al., 1983), such as those often existing around the ice sheet edges. Additional ambiguities may arise due to a dependency of the radar return signal on surface properties: In areas covered by snow and firn, the radar signal may partly penetrate the surface, and the return signal will be a mixture of surface and volume scattering. Therefore, the apparent scattering surface will depend on subsurface properties as well as the applied retracker, with the resulting elevations being biased accordingly (Michel et al., 2014; Nilsson et al., 2015). In the bare ice zone, however, the radar measures the ice surface without any elevation bias (Dall et al., 2001; Davis and Moore, 1993; Otosaka et al., 2020).

For improved data coverage and accuracy of radar altimetry in the ice sheet periphery, the SAR Interferometric Radar Altimeter (SIRAL) onboard CryoSat-2 (CS2; launched April 2010) is operated in the advanced SAR Interferometry (SARIn) mode when passing over the ice sheet marginal areas. SIRAL has two antennas, and when employed in SARIn mode, the difference in signal phase received by the two antennas is used to constrain the across-track angle of the first radar returns, whereby the origin of the echo (i.e., POCA) can be located off-nadir (Wingham et al., 2006). This approach enables the instrument to better resolve the relatively rough terrain existing at the ice sheet margin, and it partially alleviates the locational ambiguity and slope-induced errors associated with conventional low-resolution mode radar altimetry (Schröder et al., 2019). Recent developments in the processing of SIRAL SARIn data allow a wide swath of elevations to be inferred (Andersen et al., 2021; Gourmelen et al., 2018), which in the future may offer a more comprehensive view of the ice sheet topography from radar altimetry. A 1 km resolution DEM has been created for the Greenland Ice Sheet using CS2 POCA data acquired from January through December 2012 (Helm et al., 2014).

Laser-based altimeters have several advantages over radar altimeters: They have a small footprint, allowing to better resolve highly varying topography, and the laser pulse has negligible surface penetration. One drawback is, however, that data acquisition is compromised by clouds and blowing snow. The first Earth Observation laser altimeter in orbit was onboard the Ice, Cloud, and land Elevation Satellite (ICESat) (Schutz et al., 2005), from which data acquired from February 2003 through

June 2005 was used to construct a Greenland DEM at 1 km resolution (DiMarzio et al., 2007). ICESat was decommissioned in February 2010, and its successor ICESat-2 (IS2) was launched in September 2018. The ATLAS instrument onboard IS2 is a single-photon-counting lidar, providing elevation data of extremely high resolution (Markus et al., 2017). IS2 is placed in a repeat-track orbit, measuring the same ground segments with a repeat period of 91 days. An ice-sheet-wide Greenland DEM with a nominal resolution of 500 m has been constructed based on IS2 altimetry data from its first year of operation (Nov 2018 to Nov 2019) (Fan et al., 2022).

Other Greenland-wide DEMs are created primarily based on stereo-photogrammetry: Utilizing autocorrelation techniques to analyse the differences between two or more images of the same area from different angles, precise measurements of surface elevations can be obtained, allowing for the construction of elevation models in very high spatial resolution. Leveraging many thousands of overlapping pairs of sub-meter resolution optical images from the GeoEye and WorldView satellites, ArcticDEM (developed by Polar Geospatial Center, University of Minnesota) has revolutionized the availability of high-resolution elevation data for Greenland. ArcticDEM provides 'strip' DEMs covering a narrow strip of area at a specific time (Noh and Howat, 2015), as well as a mosaic providing a seamless high-resolution (2 m) comprehensive elevation model covering the entire Arctic (PGC, 2023). ArcticDEM exists in several versions, with the newest version (v4.1) (Porter et al., 2023) based on an extended data set of stereo pairs acquired during the period 2007 through 2022. Another high-resolution (30 m) stereo-photogrammetric DEM for the Greenland ice sheet is the Greenland Ice Mapping Project (GrIMP) DEM v2 (Howat et al., 2022). It is based on imagery from the same set of satellites as ArcticDEM, but using data obtained over a slightly different period (2009 through 2015) and applying a slightly different method for mosaicking the data.

The ArcticDEM mosaic is constructed by taking the median elevation value at each pixel (after removal of outliers) from the collection of strip DEMs. For improved accuracy, stereo-photogrammetry requires registration to independent elevation data. The ArcticDEM v4.1 mosaic is constructed in tiles of 100x100km, with each tile registered to the GrIMP DEM v2 (which itself is registered to IS2 elevations from the summers of 2019 and 2020 (Howat et al., 2022)), and subsequently blended at the edges to match neighbouring tiles. Edge-matching by blending and feathering of strips and tiles introduces artefacts in the final mosaic. Furthermore, areas of low radiometric contrast, and areas affected by cloud cover or shadows, may introduce errors or data voids. Other error sources include slight misregistration of the stereo-pairs, which may lead to large elevation errors in areas of high topographic variability when combining individual strips of stereo imagery to a mosaic (PGC, 2023). Despite these limitations, the accuracy and details offered by ArcticDEM have greatly improved our ability to study fine-scale topographic features across the Greenland Ice Sheet.

Satellite altimetry is collected only along the satellite tracks, resulting in a relatively sparse data set. Consequently, an altimetry-derived DEM cannot compete with a DEM based on stereographic imagery in terms of details and resolution. This is particularly true for areas with high topographic variability, such as those found in the periphery of the ice sheet, where dense sampling is required to accurately portray the topographic features of the terrain. However, to provide large-scale coverage, DEMs from stereo-photogrammetry must be based on imagery from an extended period, and thus have relatively coarse temporal resolution. While the individual ArcticDEM strips have a specific time stamp, the ArcticDEM mosaic is, for example, based on data from a 15-year period, during which the ice sheet topography may have substantially changed.

This paper focuses on the derivation and validation of PRODEMs; an annual series (2019-2022) of summer DEMs for the ice sheet marginal zone. PRODEM is the first annual series of DEMs across this rapidly changing region, directly describing the evolving summer ice topography. The PRODEMs are based on fusing CS2 and IS2 altimetry measurements acquired during the summer months, after referencing them to ArcticDEM elevations. The applied approach exploits the high spatial resolution

of ArcticDEM, while enhancing these data with a temporal resolution based on satellite altimetry. Coverage of the PRODEMs in terms of area and season is restricted to areas minimally covered by snow and firn, over which the two satellite altimeters are expected to measure the same surface, and the PRODEMs hence represent the ice surface topography. With individual

high-resolution DEMs for each summer, the PRODEM series allows detailed analyses of the annual changes in ice sheet geometry. The PRODEMs will be particularly useful for marginal mass balance considerations, as they provide crucial information on the inter-annual variability in surface elevation across these pivotal areas for understanding the complex interplay between ice sheet, oceanic processes and climate dynamics. The annual 500-meter resolution DEMs follow the naming convention PRODEMyy, with yy indicating the year, and they form part of the Programme for Monitoring of the

Greenland Ice Sheet (PROMICE) project (Ahlstrøm and the PROMICE project team, 2008).

## 2 Input data

To construct the PRODEMs, we combine summer elevation data from June through September (both months included) from two satellite altimetry missions: ESA's radar mission CryoSat-2 and NASA's lidar mission ICESat-2. Given the different nature of the sensors of the two satellite systems, their altimetry observations have distinctly different properties, including

differences in resolution and topographic sampling, as described below. We filter the satellite altimetry to only contain measurements over ice-covered areas. The observed elevations are subsequently transformed to elevation anomalies relative to ArcticDEM. All observations are referenced to the WGS84 datum, with elevations provided as heights (in meters) above the WGS84 ellipsoid.

### 2.1 CryoSat-2

The SIRAL radar onboard CS2 has monitored the elevation changes of the Greenland Ice Sheet since July 2010 (Parrinello et al., 2018). We primarily use CS2 SARIn altimetry from the ice sheet margin. In SARIn mode, SIRAL has a footprint of ~400 m along track and 1.65 km across track, corresponding to a total footprint area of 0.5 km$^2$ (ESA, 2021). For a small part of the PRODEM area in South West Greenland, CS2 is operated in Low Resolution Mode (LRM) instead of SARIn mode, and in this small area, we include LRM CS2 altimetry for generating the PRODEMs.

We use CS2 Ice Level 2 Baseline E data (ESA, 2023), and remove data flagged as having issues with accurate elevation retrieval and/or cross-track angle error (ESA, 2021), and hence geolocation error. Additionally, observations without relocation from nadir or with relocation distances exceeding 15 km are considered unrealistic, and these observations are also removed. On average, 23-24 % of the original data set is discarded, mostly due to cross-track angle errors.

### 2.2 ICESat-2

With the launch of IS2 in September 2018, very high-resolution laser altimetry data for the Greenland Ice Sheet has become available. Each IS2 track consists of six beams separated into three pairs, with approximately 3 km between pairs, and 90 m between adjacent beams within a pair (Markus et al., 2017). Each pair consists of one strong and one weak beam, and given the short distance between them, the measured elevations along the two beams are strongly correlated. Dense sampling in

certain areas may, however, lead to overfitting of local variations, thereby introducing biases or inaccuracies in the interpolated elevation fields. To avoid excessive oversampling along the beams, the PRODEMs are constructed using data only from the three strong beams.

We use the IS2 Land Ice Height data set ATL06 V6 (Smith et al., 2023), which is a down-sampled product where individual

photon heights are averaged within 20 m segments along each beam (Smith et al., 2019). Observations flagged as bad data are

removed (3-7 % depending on year). For consistency with the resolution of the CS2 observations, we further down-sample the ATL06 data by computing median values over 250 m along-track segments of each beam. Segments containing less than five elevation values (i.e. less than 1/3) are discarded from the analysis. After down-sampling, the comparability with the CS2 elevation data is increased, also in terms of the relative number of observations from the two satellite missions included in the analysis. IS2 observations constitute 71-76 % of the total ice elevation measurements used for generating the PRODEMs.

## 2.3 ArcticDEM

All altimetry measurements are transformed to elevation anomalies relative to a reference DEM, for which we employ the 500 m resolution ArcticDEM v4.1 gridded mosaic (Porter et al., 2023). ArcticDEM is also employed for providing an estimate of the varying surface roughness, which is used for computing the spatial component of the observation uncertainties (Section 4.1). A map of the 100-meter scale surface roughness is obtained from the 100 m resolution ArcticDEM mosaic, computed as the largest elevation difference between the central pixel and its surrounding cells. We note that with this definition a flat sloping surface will have a roughness value depending on the slope.

## 3 PRODEM: Coverage and resolution

### 3.1. Spatial and temporal coverage

The PRODEMs are summer DEMs for the marginal areas of the Greenland Ice Sheet, including peripheral glaciers connected to the ice sheet based on the BedMachine v5 ice cover mask (Morlighem et al., 2017, 2022). Each PRODEM in the series is independent. They are built entirely from summer data (June 1[st] to September 30[th]), and the DEM coverage is limited to the outermost 50 km wide band of the ice sheet margin. To avoid edge effects in the DEMs, altimetry is incorporated from an extended area that encompasses a 10 km inland buffer zone. The PRODEM area covers most (~90 %) of the summer bare ice zone in recent years (Fausto and the PROMICE team, 2018). The area and period are selected to ensure that the altimetry data primarily provides snow-free ice surface elevations, thereby largely eliminating the effect of different snow penetration depths of the radar versus the laser signal.

With the collected data from CS2 and IS2 combined, the data coverage is sufficient to derive high-resolution annual maps of ice sheet topography in the marginal areas of the Greenland Ice Sheet. Using 500 m grid resolution, the interpolation distances from the grid cell centres to the nearest datapoint show a distribution with a mode in the 120-230 m range, and a median of 452 m (values for 2019 data; see histogram Fig. 1a). Due to the denser satellite orbits towards the north, the interpolation distances tend to be smaller there. Within any region, the data coverage is irregularly spaced, and in a minority of cases, the accuracy of the interpolated elevations is significantly limited by data availability. This is especially an issue for the southernmost drainage basin (basin 5). Across all PRODEM grid cells, the closest point is more than 1 km away ~15 % of the time, but further away than 2 km only ~3 % of the time. The majority of cells (~85 %) have a sample point less than 1 km away (Fig. 1a).

Figure 1b-d shows the 2019 data coverage in the regions around the three largest outlet glaciers from the Greenland Ice Sheet: Sermeq Kujalleq (Jakobshavn Isbræ), Kangerlussuaq Glacier and Helheim Glacier. The observations are distributed differently in other years. Areas of lower data density will be reflected as regions of increased uncertainty in the derived PRODEM elevations for the given year, which are noticeable from the accompanying map of elevation field uncertainty provided with each PRODEM.

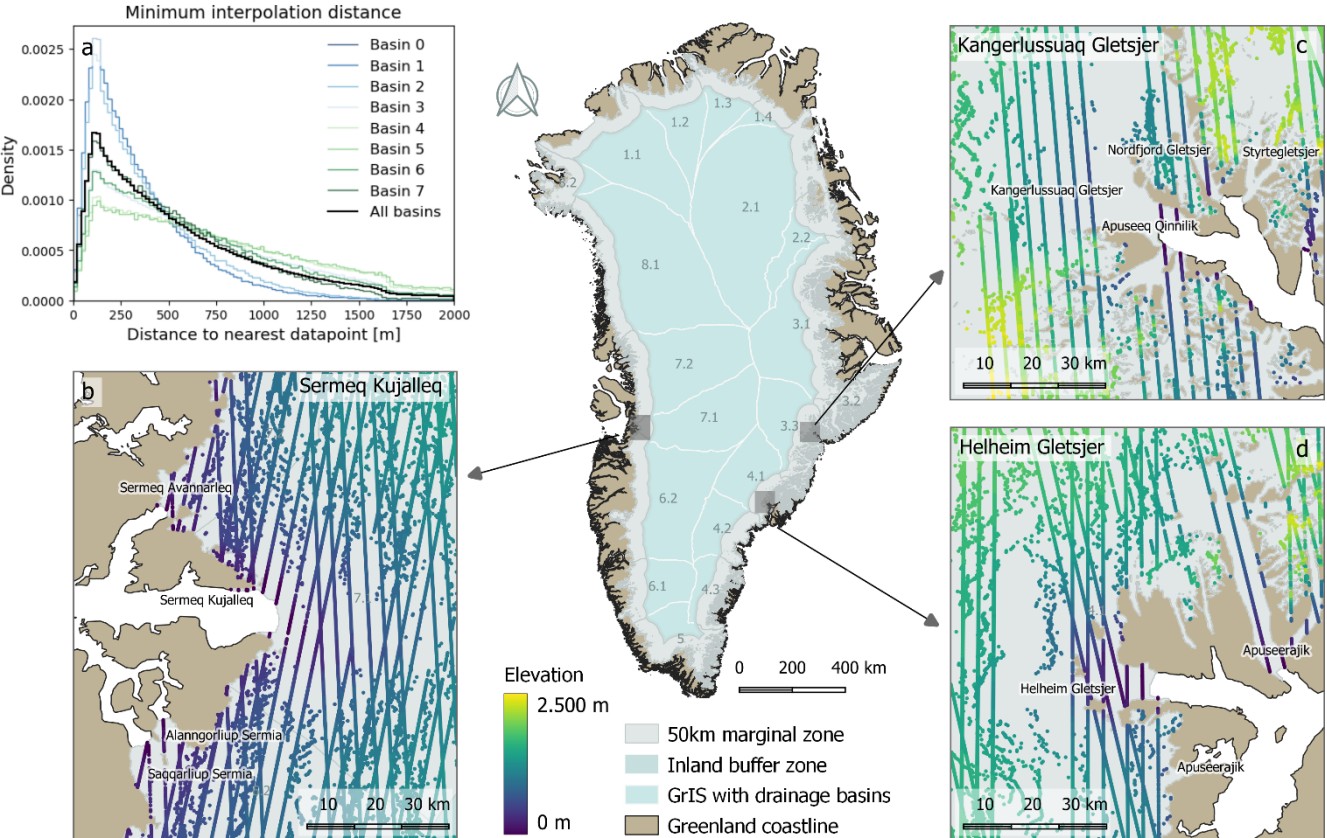

**Figure 1: a):** Histogram showing the distribution of distances for each grid cell centre to the closest observation (data from 2019). Distributions are shown for each main drainage basin (blue colours for drainage basins in North and East Greenland; green colours for basins in South and West Greenland; colour saturation is relative to basin latitude), and for the entire PRODEM area (black). **b-d):** CS2 and IS2 data coverage for June through September 2019 over subsectors covering the area around the three largest outlet glaciers from the Greenland Ice Sheet: Sermeq Kujalleq (b), Kangerlussuaq (c) and Helheim Glaciers (d). IS2 data can be recognized as data acquired along straight lines, while CS2 data is geolocated according to the POCA. For the derivation of the PRODEMs, the ice sheet marginal zone was first divided up into subsectors using the drainage basin definitions from Zwally et al (2012), which are indicated with numbers on the overview map of the Greenland Ice Sheet. Official glacier names from (Bjørk et al., 2015).

### 3.2. Spatial resolution

The PRODEMs are constructed by evaluating the interpolated elevation anomaly field at the grid cell centres (an approach called 'Point Kriging' (Hengl, 2009)). For this approach to accurately represent the mean field value within a grid cell, the grid resolution must align with the scale of variability of the observations. After down sampling of the IS2 data, both CS2 and IS2 data sets are representative for an area of a few hundred meters, and an appropriate resolution for the anomaly field of the interpolated PRODEMs is therefore on the order of 500 m.

### 4 Estimating uncertainties and bias of the satellite altimetry data

### 4.1 Observation uncertainties

During the DEM construction based on the CS2 and IS2 satellite altimetry, we include a measure for the uncertainty of the individual data points to inform how much a given measurement should constrain the interpolated field. Proper uncertainty estimation is important: If uncertainties are overestimated, the resulting field will be too smooth, thus discarding much of the information existing within the observations. Conversely, if uncertainties are underestimated, the interpolation will tend to overfit the observations while inducing noise in the interpolated field.

The observation uncertainty should be understood as a measure of how well an observation represents the mean elevation field (spatial and timewise) at a given location. The degree to which an observation represents the mean elevation field may be

divided into four factors, which together constitute the total uncertainty: The instrument measurement uncertainty ($\sigma_{meas}$), the elevation uncertainty caused by potential errors in the geolocation of the observation ($\sigma_{geo}$), along with the spatial ($\sigma_{spatial}$) and temporal ($\sigma_{temp}$) representability of an observation. Appropriate values for the latter two depend on the elevation field to be interpolated. For a 500-m resolution DEM, $\sigma_{spatial}$ is a measure of how well an observation represents the average elevation field within a similar range, which depends on the spatial variability of the local elevation field. Further, the observations are gathered within a 4-month window, during which the surface elevation is changing over time, and we are to obtain a reconciled mid-summer elevation field. The term $\sigma_{temp}$ indicates how well a given observation represents the mid-summer elevation value at its location. The total observation uncertainty may be assessed by combining the uncertainty components in quadrature under the assumption that they are independent.

Given the distinct nature of the CS2 and IS2 sensors, the importance of the various uncertainty components differs for the two sets of observations. In the following, we develop separate uncertainty models for the two data sets to estimate representative values for their spatial uncertainty (including effects from geolocation and instrument measurement uncertainty), and a common model for the temporal uncertainty component.

**4.1.1 Spatial uncertainty of the CS2 elevations**

When operating in SARIn mode, the CS2 SIRAL radar system provides one measurement of the surface elevation within the satellite footprint of 0.5 km². The slope-induced errors associated with the SARIn data are much smaller than for conventional low-resolution mode radar altimetry, but may still be present on smaller glaciers and ice caps as well as in the ice sheet marginal areas, such as those in this study, where the topographic relief may be significant (Wang et al., 2015). Furthermore, the geolocation of the measurement (POCA) is associated with some uncertainty due to phase ambiguities, which may give rise to elevation outliers.

To estimate the spatial uncertainty associated with the CS2 observations, we analyse the measured elevation differences at intersecting CS2 satellite tracks within our study area. To mitigate the impact of temporal variability, the time interval between the two measurement acquisitions is restricted to be less than 15 days. For each observation in a satellite track, the closest observation from another track within a 50 m search radius is selected, and their associated elevations are compared. Prior to comparison, the elevations are adjusted relative to ArcticDEM to account for any general slope in the area. The observed elevation differences are primarily a result of surface fluctuations between the two adjacent measurement locations caused by small-scale topographic variability, although short-term elevation changes due to e.g. intermittent snowfall may also play a role.

A total of 5873 close-in-time intersecting CS2 tracks exist within our study area during the four summer periods (2019-2022). The resulting distribution of observed elevation differences near the track intersections is sharply peaked around 0 m (Fig. 2a; grey histogram) with a median absolute deviation (MAD) of 0.86 m. While not strictly adhering to a Gaussian distribution, an approximate standard deviation describing the spread of the distribution can be obtained using normalized MAD: $nMAD = 1.4826 \cdot MAD$. This approach has the advantage of being more robust to outliers than a direct calculation of the standard deviation. As the distribution of elevation differences includes the error sources from both elevation measurements, an approximate measure of the 1σ uncertainty of the CS2 observations can then be derived as:

$$\sigma_{spatial}^{CS2} \approx \frac{nMAD}{\sqrt{2}} = \frac{1.4826}{\sqrt{2}} \cdot MAD \tag{1}$$

Using this equation, the mean uncertainty of all CS2 observations due to spatial variability of the terrain is found to be 0.90 m.

It is not a very good approximation to apply a constant spatial uncertainty to all CS2 observations. Several factors are expected to cause a larger uncertainty in areas of sloping and/or rough terrain: The spatial representativeness for any given observation will be smaller, and an error caused by incorrect geolocation will lead to a much larger error in the obtained elevation estimate.

We therefore investigate the relationship between local 100-meter scale surface roughness (obtained from ArcticDEM) and the dispersion of measured elevation differences near intersecting satellite tracks. Binning the satellite crossings according to the local roughness, statistics for the distributions of measured elevation differences are computed for different roughness values. The observations are divided into 20 roughness bins, each containing a sufficiently large number (~280) of observations to obtain good estimates for MAD, with extreme roughness values (above the 95th percentile) excluded due to poorly

constrained statistics. A linear relationship exists between the logarithm to the local 100-meter scale roughness and $\sigma_{spatial}^{CS2}$ (Fig. 2b), while plateauing at a value of 0.39 m for low roughness values (log(roughness) < 0, i.e. roughness < 1 m). At these locations with little slope and surface irregularity, the total observation uncertainty is dominated by the measurement uncertainty, and we thus evaluate the instrument measurement uncertainty ($\sigma_{meas}$) of the CS2 observations to be ~40 cm.

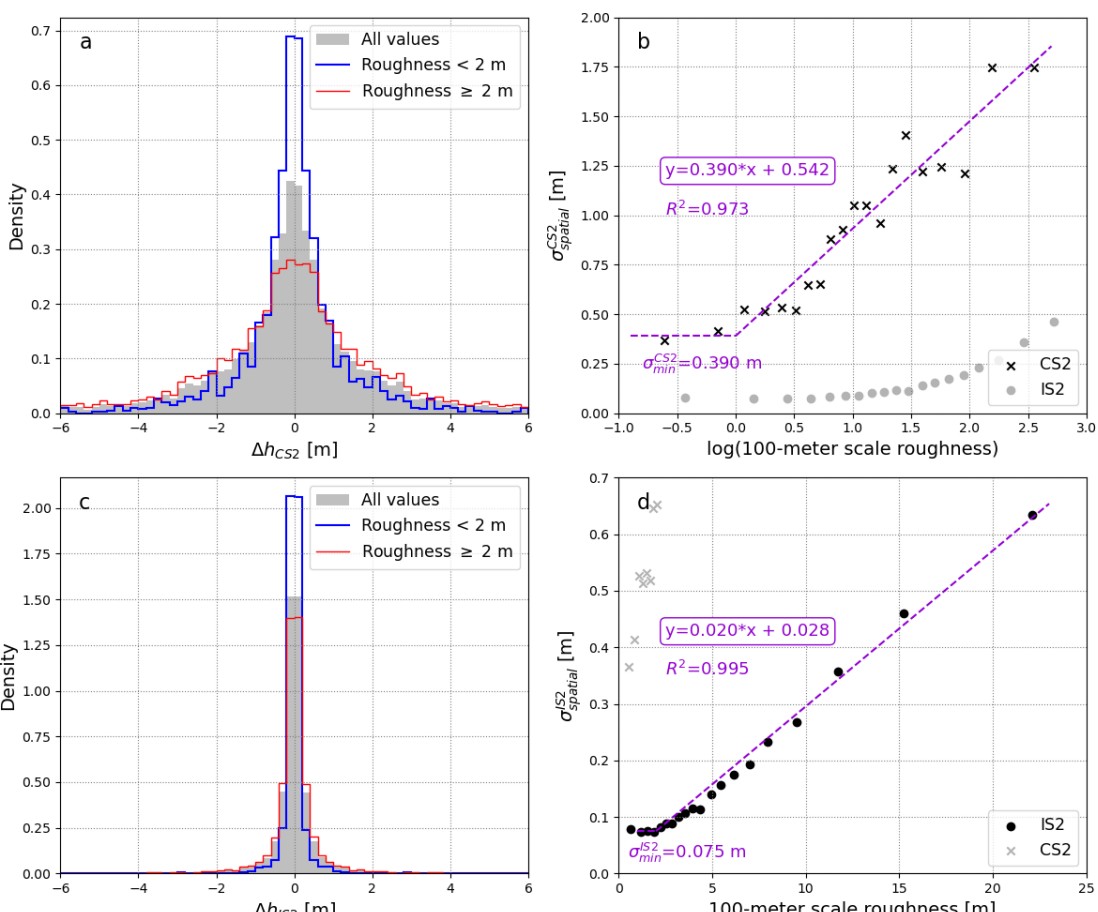

**Figure 2: a, c) Distribution of measured elevation differences ($\Delta h$) close to CS2 (a) and IS2 (c) intersecting satellite tracks within the entire study area (grey), and from intersecting tracks in areas with roughness higher (red) or lower (blue) than 2 m, respectively. Only data from temporally close acquisitions is included. b, d) For CS2 (b), a linear relationship (dashed violet line) exists between the logarithm to the local 100m-scale roughness and the estimated spatial uncertainty based on the dispersion of associated elevation differences (Eq. 1). For IS2 (d), on the other hand, we observe a linear correlation directly between the roughness and estimated**

**spatial uncertainty. To ease the comparison between the derived CS2 and IS2 uncertainties, data from the other satellite is also included in b and d (light grey). All uncertainty models are based on data from the summers of 2019 through 2022.**

Spatial differences in CS2 observation uncertainties due to surface roughness are estimated accordingly: The logarithm to the local 100-meter-scale roughness is computed, and we apply the obtained linear relationship (Fig. 2b) while imposing a minimum uncertainty value of 0.39 m in areas of small surface roughness (< 1 m). The resulting distribution of the spatial

uncertainty for the CS2 observations within our study area has a median value of 0.96 m (Fig. 3a; blue histogram). Reassuringly, this is very similar to the previously quoted value of $\sigma_{spatial}^{CS2}$ (0.90 m) calculated based on the dispersion of elevation differences at all temporally-close intersecting satellite tracks.

Lastly, it should be noted that the CS2 observations are inherently biased towards higher elevation values: Since the elevation
estimates from CS2 are associated with the POCA within the satellite footprint, CS2 will preferentially sample the topographic highs rather than topographic lows located within the footprint. This inherent measurement bias of the CS2 elevations is not accounted for. One consequence of this bias is that the measured elevation differences obtained near the intersection of two satellite tracks may not reflect the full variability of the terrain over small distances. This will result in an underestimation of the observation uncertainty due to spatial variability, particularly in highly irregular terrain, and it is a likely cause behind the
diminishing rate of increase in uncertainty at high roughness values.

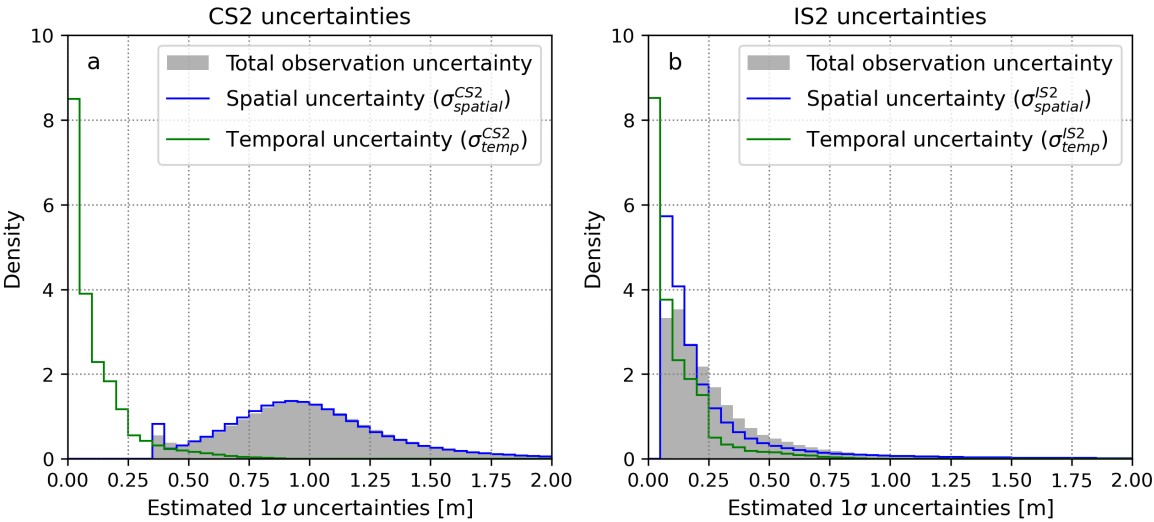

Figure 3: Distribution of derived total uncertainties (grey), along with the individual spatial (blue) and temporal (green) uncertainty components for the CS2 (a) and IS2 (b) data, respectively. Data from summers 2019 through 2022.

**4.1.2 Spatial uncertainty of the IS2 elevations**

The measurement precision of the IS2 ATL06 product has been documented to be as small as 9 cm over the flat interior part of Antarctica (Brunt et al., 2019), and, compared to CS2, the ATLAS instrument provides much better resolved and localized elevation data with no topographic preference (Magruder et al., 2020). Hence, apart from outliers due to occasional errors in the ATL06 data caused by errors in tracking the surface within the photon cloud, the total observation uncertainty is dominated by the spatial and temporal variability of the surface elevation relative to the mean elevation field.

By construction, the 250-meter median-averaged IS2 data used for DEM construction (section 2.2) already includes spatial averaging in the along-track dimension. However, spatial representability of the along-track-averaged values depends on the local topography; the precise location of the track will influence the resulting average more in areas of high topographic relief. Applying a similar approach as for the CS2 observations, the uncertainty contribution from spatial representability of the IS2
observations is estimated by analysing the dispersion of measured elevation differences at close-in-time (acquired within 15 days) adjacent observations (located within 50 m from each other) obtained near intersecting satellite tracks across various ArcticDEM roughness intervals. Based on data from a total of 12169 track intersections binned into 20 roughness intervals, we observe distinct differences to the derived CS2 uncertainties (see section 4.1.1): For the IS2 data, the spread of the measured elevation differences is much smaller (Fig. 2c), and the spatial uncertainty increases linearly with the terrain roughness (Fig.
2d). At low roughness values (< 2 m), the uncertainty stabilizes at a 1σ value of ~8 cm, consistent with the previously reported instrument measurement uncertainty for flat terrain (Brunt et al., 2019).

The distribution of the individually estimated IS2 spatial uncertainties (Fig. 3b; blue histogram) has a median of 0.16 m. We also here note the excellent agreement with the value (0.15 m) estimated based on the spread of the elevation difference distribution of all intersecting IS2 tracks within the study area.

### 4.1.3    Uncertainty due to temporal variability

During the four-month observational period used for construction of each PRODEM, the ice sheet surface topography evolves due to surface mass balance and/or dynamic changes taking place over the summer. The rate of elevation change during summer is highly variable across the ice sheet. Generally, the margins of the ice sheet experience the largest elevation changes, albeit with considerable regional variability in magnitude (Slater et al., 2021b). To account for these changes in elevation during the observation acquisition period, a temporal uncertainty component is assigned to each of the observations based on its proximity to mid-summer ($t_0$), which we take to be the middle of the observation period, i.e. August 1$^{st}$.

To assign a temporal uncertainty component to the CS2 and IS2 elevations, we take advantage of a dataset of average summer (May-August) elevation trends over the period 2011 through 2020 derived from CS2 altimetry (Slater et al., 2021a). We approximate $\sigma_{temp}$ as the absolute value of the product of the local long-term average rate of summer elevation change and the time difference between acquisition and mid-summer:

$$\sigma_{temp} = \left| \left( \frac{dh}{dt} \right)_{summer} \cdot (t - t_0) \right| \tag{2}$$

In areas not covered by data for average summer elevation trends, the average thinning of the ablation zone during the 2011-2020 period of 1.4 m/year (Slater et al., 2021) is used.

The resulting distributions of temporal uncertainties for the individual CS2 and IS2 observations are illustrated in Fig. 3 (green histograms). Both distributions have a median of 0.07 m. In general, for both instruments, the temporal uncertainty is notably less important than the spatial uncertainty component, particularly evident in the case of CS2. However, for individual observations the temporal uncertainty component may be non-negligible, with 21 % of the IS2 observations having larger temporal than spatial uncertainty. For CS2, this is only the case for 0.4 % of the observations.

### 4.1.4 Combined observation uncertainty

To obtain a measure for the total observation uncertainty, the spatial and temporal uncertainty components are combined in quadrature under the assumption that they are independent (Fig. 3, grey histograms). The assumption of independence is partially justified: While both surface roughness and summer elevation trends tend to increase towards the ice sheet margin, the temporal uncertainty estimate is strongly influenced by the acquisition time relative to mid-summer, which largely removes the correlation between the two uncertainty components.

The derived observation uncertainties are subsequently tested for consistency with the observed elevation differences near intersecting satellite tracks. Computing z-scores from the observed elevation differences by dividing these with their estimated combined uncertainty, the resulting distributions are found to adequately approximate standard normal distributions, albeit heavier tails indicate the existence of outliers not fully captured by the estimated uncertainties.

The distributions of total observation uncertainty have a median value of 0.98 m (CS2) and 0.21 m (IS2), respectively. The derived uncertainties display large spatial variability, with much larger uncertainty values towards the margin where the

topography tends to be substantially rougher than in the smoother, and more temporally stable, inner part of the ice sheet marginal zone.

## 4.2 Bias estimation

As previously mentioned, we expect a disparity between elevation measurements obtained from CS2 and IS2 over firn and
snow-covered surfaces due to differences in penetration. This may give rise to biases, which, if existing, must be corrected before combining elevation data from the two satellites. The PRODEMs, however, are constructed based on altimetry from the ice-sheet marginal zone during summer, where the snow and firn cover is limited, and we therefore expect small biases only between the employed CS2 and IS2 altimetry within the PRODEM area.

This assumption is verified by an analysis of IS2 versus CS2 elevations at intersecting satellite tracks, while imposing restrictive limits of a maximum distance of 50 m and a maximum time difference of 15 days between the two acquisitions. We find no evident bias between the elevation estimates obtained from the two satellite sensors: Based on observations from all 8905 intersecting tracks from summers 2019-22 fulfilling these criteria, the elevation differences are found to be distributed with a median value close to zero. The median of the distribution tends to be smallest in magnitude for areas of low roughness
(-0.05 m; Fig. 4a), where the elevation differences are best determined and the dispersion of the elevation difference distribution is small (Fig. 4b). The negative value implies that CS2 elevations overall tend to be slightly lower than the IS2 elevations, although this is not consistent across drainage basins. We consider this value (-0.05 m) to be an upper limit for the potential bias between the two altimeters over bare ice, since areas of flat terrain are predominantly located in the high-elevation inner part of the PRODEM region, which may periodically be covered by snow during the designated summer period.
This upper limit for a potential bias is well within the uncertainties of the CS2 observations (Fig. 3a). We can therefore conclude that, in accordance with earlier studies (Dall et al., 2001; Davis and Moore, 1993; Otosaka et al., 2020), we do not observe any significant elevation bias between the two sets of altimetry in this region within their associated uncertainties.

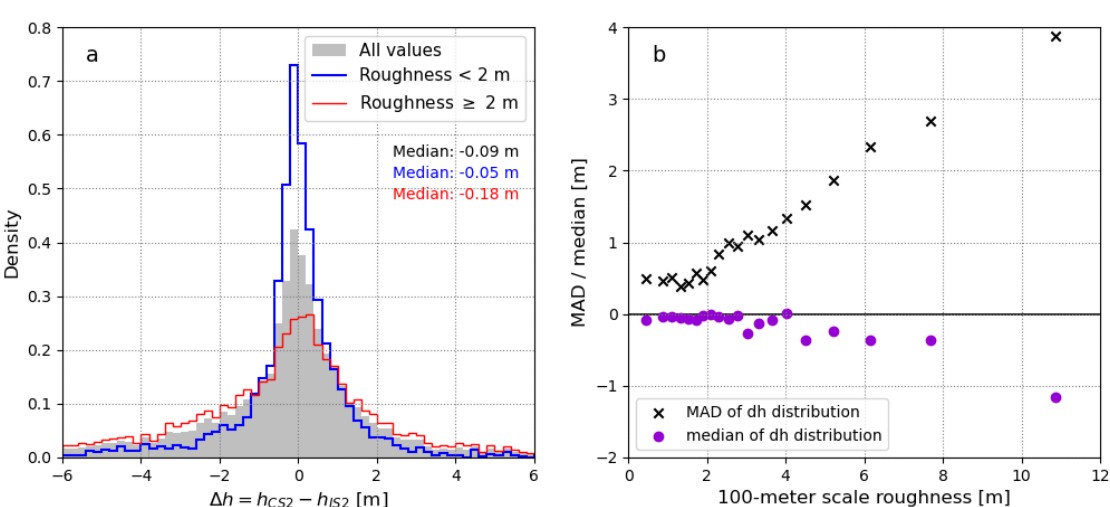

**Figure 4: a) Distribution of elevation differences, computed as $\Delta h = h_{cs2} - h_{is2}$, near intersecting CS2 and IS2 satellite tracks, when**
**imposing strict limits in terms of maximum distance and time between acquisitions. b) Evolution of the median (violet dots) and MAD (black crosses) of the resulting elevation difference distribution as function of the 100-meter-scale roughness of the terrain. Data from the PRODEM area during summers 2019 through 2022.**

## 5 Constructing the PRODEMs

### 5.1 Regional PRODEMs for marginal drainage basin subsectors

For computational purposes, we create regional PRODEMs for marginal subsectors based on drainage basin outlines, which are subsequently combined to form a full-margin DEM. We use the drainage basin definitions from Zwally et al. (2012), which divide the Greenland Ice Sheet into 19 drainage basins (Fig. 1). For each sub-sector DEM, data from a 10 km neighbourhood is included to avoid artefacts around drainage basin edges.

### 5.2 Elevation anomalies relative to ArcticDEM

Prior to interpolation, the satellite altimetry is detrended by subtracting a reference DEM (ArcticDEM v4.1) from each elevation measurement using linear interpolation. The resulting data set is labelled elevation anomalies. The elevation anomalies display a relatively smooth field valid for interpolation (Fig. 5). Even for areas with very rough terrain, such as those often found at the very outer part of the ice sheet edges, the observed elevation anomalies are consistent across length scales of 1-2 km (Fig. 5b).

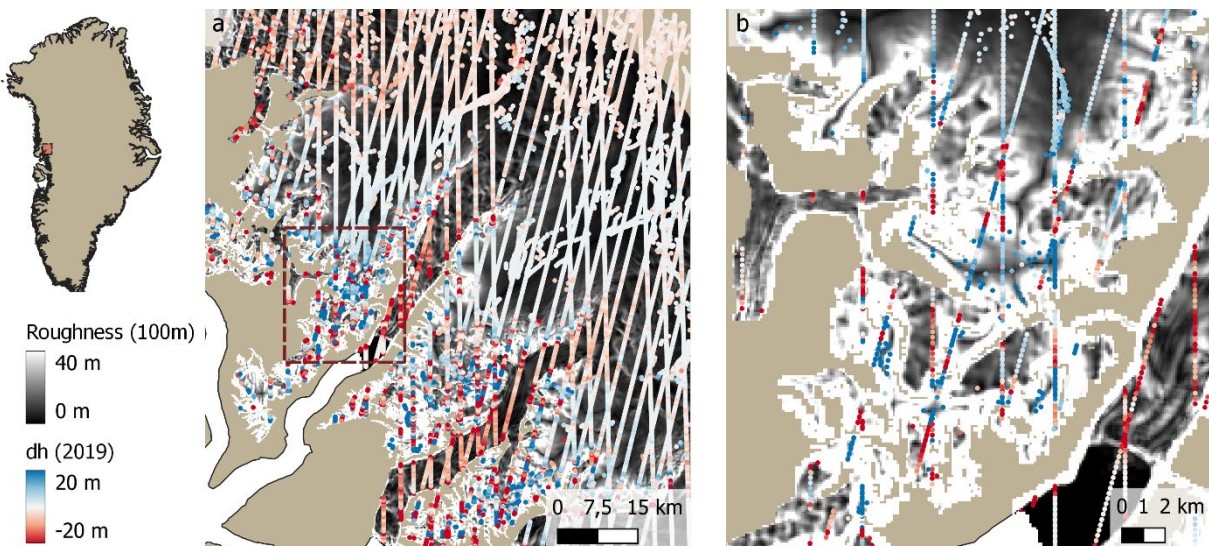


**Figure 5: Satellite altimetry from CS2 and IS2 after calculating the elevation anomalies to ArcticDEM for a marginal area in Central West Greenland (red square in the overview map) with significant topographic relief. a) The elevation anomalies display larger variability in the outer regions with the highest roughness (white areas), b) but a zoom-in to an area with extreme topographic relief (square marked with a red dashed line in a) shows that even in these areas, the elevation anomalies are consistent across length**
**scales of 1-2 km. IS2 observations can be recognized as data acquired along straight lines, while CS2 observations are geolocated according to the POCA.**

Based on the observed elevation anomalies relative to ArcticDEM, some outliers in the altimetry are clearly identifiable. The data is cleaned by successively applying a 10x10 km spatial filter to the elevation anomaly data. Observations outside the 5σ range of local variability are identified as outliers and removed. The filtering is repeated 10 times, after which no or few outliers are detected. During the construction of PRODEM19, a total of 3.1 % of the CS2 data and 0.2 % of the IS2 data (together constituting 1.0 % of all data) are flagged as outliers. Further investigations show that many IS2 observations flagged as outliers are likely not measurement errors. The vast majority of these are located next to a nunatak, and the irregularity in elevation anomalies in these areas, as picked up by the spatial filter, is caused by large variability in the reference field obtained by linear interpolation of the 500-meter resolution ArcticDEM to the measurement locations. We therefore attribute this
perceived irregularity to resolution issues in the applied ArcticDEM mosaic. In contrast, the CS2 observations flagged as outliers are evenly distributed throughout the study area, suggesting that these indeed represent inaccurate altimetry measurements, as caused by e.g., incorrect geolocation of the observations.

The PRODEMs are constructed using the same projection (EPSG:3413) and grid nodes as the ArcticDEM 500 m mosaic, thus avoiding additional interpolation when later adding the interpolated map of elevation anomalies to the reference DEM.

## 5.3 Spatial interpolation of the elevation anomaly field

The PRODEMs are spatially interpolated from the satellite altimetry point measurements by Ordinary Kriging (see e.g. Hengl (2009)). In essence, Kriging performs an optimal unbiased linear prediction of a continuous field by exploiting knowledge of the spatial covariance of the field. The spatial covariance is specified in terms of a variogram, which can be extracted from the point cloud, and this information is subsequently used to assign distance-dependent weights to nearby data points. Several Kriging variants exist. In Ordinary Kriging, the predictions are formed as the sum of a locally constant function and a spatially correlated stochastic field. Kriging can account for variable uncertainty of individual data points during the interpolation, and it is a widely used method to interpolate geophysical fields due to its robustness in capturing spatial variability (e.g. (Bales et al., 2001; Bamber et al., 2001b; MacGregor et al., 2015).

Ordinary Kriging relies on the data to be isotropic (i.e. have non-directional properties) and have second-order stationarity (i.e. with constant mean and variance fields) so that the spatial covariance structure is representative across the study space. While it is reasonable to assume the elevation field to be isotropic, neither the ice surface elevation field nor the elevation anomaly field relative to ArcticDEM are second-order stationary. Ice sheet surface elevations strongly violate the stationarity requirements, with mean elevations decreasing considerably towards the ice sheet margin, whereas the transformed data set of elevation anomalies does not show a similar trend. For neither data set, however, the field variance is stable across space: Both the elevation and the elevation anomaly fields generally display higher variance in the topographically rough areas near the ice sheet margin than in the smoother inland areas.

Consequently, a single variogram cannot properly describe the spatial covariance of the elevation anomaly field relative to ArcticDEM. To account for this, we first estimate annual maps of the variogram parameters describing the varying spatial correlation structure of the elevation anomalies across the PRODEM area (section 5.3.1). For a given location in the PRODEM grid, an appropriate set of variogram parameters is subsequently extracted and applied in the Kriging routine (section 5.3.2) to produce an interpolated value for the elevation anomaly. The final PRODEM is constructed by adding the map of derived elevation anomalies to the ArcticDEM mosaic, thereby forming an updated summer DEM for the region. Under the assumption of Gaussian uncertainties on the input data, Kriging can further employ the field's spatial characteristics to assess the uncertainty of the interpolated elevation field. The accuracy of the interpolated field will be limited if the number of sampled observations within the effective interpolation radius is small, or if the sampled observations are inadequately distributed relative to the spatial properties of the field. Accuracy of the PRODEMs will therefore be reduced in areas of high variability and few observations, such as the conditions prevalent at the outer edges of the ice sheet.

### 5.3.1 Modelling the regionally-varying spatial covariance structure

We first assess the varying spatial covariance structure of the elevation anomalies across the 50 km marginal ice sheet zone forming the PRODEM area. Spatial covariance may be represented by a variogram, which is a measure of half the variance of the differences in field value at two locations as function of the distance ("lag") between those locations. To account for the spatially-varying covariance structure of the elevation anomaly field, individual variograms are constructed for points in a 10 km grid covering the PRODEM area. The empirical variograms are computed using the Dowd estimator (Dowd, 1984; Mälicke et al., 2022), which is robust to extreme outliers.

From the point cloud of elevation anomalies, we observe that substantial changes in field characteristics sometimes take place over small spatial scales (<5 km). Such rapid change occurs, for instance, in regions where a smooth low-elevation outlet glacier is surrounded by higher elevation ice-covered mountainous areas, with the latter displaying substantially higher spatial variability also in the elevation anomalies. Nevertheless, to provide sufficient data for variogram estimation, we assume stationarity within a slightly larger area: Each variogram is based on the nearest 2000 observations within a 30 km radius of the grid cell centre, the first condition generally being the most restrictive (mean distance to the observation furthest away is 20 km). We use 200 m binning of distance lags and a maximum bin size of 10 km, which generally ensures sufficient data for consistent retrieval of the variogram at all lags.

The empirical variograms are subsequently modelled using the exponential covariance function (Hengl, 2009)Click or tap here to enter text.. This is a widely used covariance model suitable for continuous spatial fields with relatively rapid changes and localized features, and it provides a good description of the various covariance structures existing within the elevation anomaly field. The exponential covariance function is given as:

$$C(d) = \sigma^2 \exp\left(-\frac{d}{\rho}\right) \tag{3}$$

Here, $C(d)$ is the covariance between two points separated by distance $d$, $\sigma^2$ is the variance of the field, and $\rho$ is a length scale parameter. The latter is a measure for how far an observation carries information on the surrounding field; the covariance of two points spaced with a distance of $\rho$ is approximately one third of the field variance. We define the effective range, $L_e$, as the distance after which two data points are essentially no longer correlated (covariance has decreased by 95 %), which can be calculated as $L_e \approx 3\rho$. For infinitesimal separation distances, a nugget effect ($\sigma_n^2$) may be added to the covariance function. The nugget represents measurement uncertainties and/or micro-scale variability in the data, e.g., natural variability at scales smaller than the sampling distance, both of which will cause two measurements at essentially the same location to differ slightly. In our case, a nugget will be present due to observation uncertainties, and we therefore prescribe the nugget based on the squared median uncertainty of the surrounding observations. Adding a nugget effect, a model for the variogram, $\gamma(d)$, can be obtained from the covariance function by (see e.g. Hengl (2009)):

$$\gamma(d) = \sigma^2 + \sigma_n^2 - C(d) \tag{4}$$

An exponential variogram model is fitted using robust least squares to each of the empirical variograms in the coarse-resolution grid across the PRODEM area. The length scale, $\rho$, is constrained to the interval 500 m to 20 km (effective range: 1.5 - 60 km). No bounds are imposed for the variance. The predicted field values will tend to be dominated by many nearby observations carrying high weights in the interpolation, and it is therefore most important to correctly capture the covariance structure at close distances (small lags). To ensure that the fitted variogram models are well representing the covariance at small lags, these are weighted higher in the fitting procedure: Applied weights are inversely proportional to the lag value of the empirical variogram. We further adjust the weights according to the number of observations at each lag, such that semi-variances calculated based on few observations are weighted less. With the general distribution of observations, this leads to approximately 50 % weighting at 5 km. Figure 6d-e shows four examples of representative empirical variograms and fitted variogram models. The procedure is repeated for each year in the PRODEM series. In the following, we focus on the variogram parameter fields obtained from summer 2019 data, but the overall patterns are consistent across the four years currently covered by the PRODEM series.

To avoid discontinuities in the final PRODEMs, the variogram parameter fields must be spatially continuous. Continuity of the variogram parameters is also expected since adjacent models are based on partly the same data set; the variograms are obtained for each point in the 10 km grid, with each variogram based on observations within a 30 km radius. To ensure continuity, the parameter fields are smoothed using a 3x3 pixel (i.e., 30x30 km) spatial median filter.

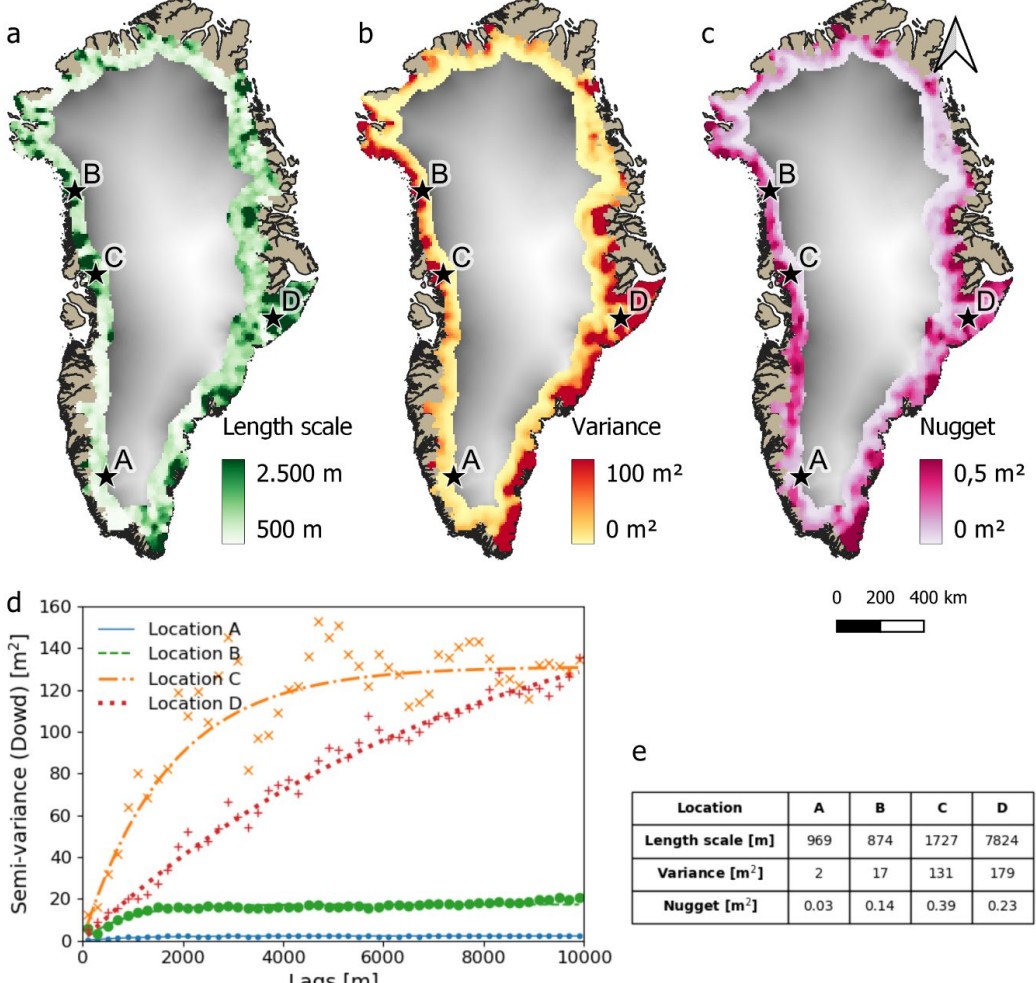

**Figure 6: a, b, c) Spatial variability across the PRODEM area of the three parameters of the exponential variogram model; length scale (a), variance (b) and nugget (c), respectively. The filtered parameters fields for 2019 data are shown. ArcticDEM ice surface elevations (greyscale) are used for background. d) Four representative empirical and fitted variogram models found within the PRODEM area; locations are indicated with letters A-D in the maps. These are selected based on a K-means cluster analysis of the variogram parameters and selecting the variograms most representative for each cluster centre. e) Table of variogram parameters corresponding to the modelled variograms shown in d.**

The large-scale patterns of variogram parameters (Fig. 6a-c) reflect the differences in spatial covariance across the marginal ice sheet zone. Elevation anomalies in the innermost part of the marginal zone are generally characterized by smooth fields, with low variance and small nugget values (Fig. 6, location A), implying that the elevation anomaly in these areas is very well-characterized by nearby observations. The variograms here tend to display relatively small length scales, reflecting that local peaks and valleys exist within the otherwise smooth anomaly field. Moving towards the margin, the variograms gradually change towards higher variance and nugget values, but the length scale often remains small (Fig. 6, location B). Upstream outlet glaciers, we observe a varying pattern of high variance and nugget values, often combined with larger length scales (Fig. 6, locations C and D). Particularly East Greenland outlet glaciers tend to be characterized by extremely high variance (location D).

For each PRODEM 500 m grid cell, an appropriate set of variogram model parameters is found by bilinear interpolation in the coarser parameter grid, and these are used as input to the kriging routine.

**5.3.2 Interpolated fields of elevation anomalies and associated uncertainty**

For each grid cell, the Ordinary Kriging interpolation is based on up to 200 nearby observations. The input data, however, is not distributed evenly, particularly given the closely spaced IS2 observations obtained along straight lines. To ensure that data

included in the interpolation is well distributed, these data are selected by dividing the neighbourhood around the grid cell centre into 8 subsectors and using the nearest 25 observations from within each subsector.

The expected field value at location $x_0$, here denoted $\hat{f}(x_0)$, can be calculated by weighting the surrounding $J$ observations $(x_i, z_i)$ according to their distance to $x_0$ using the local variogram function:

$$\hat{f}(x_0) = \mu + \boldsymbol{\lambda}_0^T \cdot (\boldsymbol{z} - \mu) \tag{5}$$

Here, $\boldsymbol{z}$ is a vector with the field observations, and $\boldsymbol{\lambda_0}$ is a vector containing the weights for these at location $x_0$. We apply the local median of the observations as a robust estimator for the local mean value, $\mu$, of the elevation anomaly field. The weights can be calculated by the following matrix equation (Paciorek, 2008):

$$\boldsymbol{\lambda_0} = (\boldsymbol{C} + \boldsymbol{N})^{-1} \boldsymbol{c_0},$$

where $\boldsymbol{C}$ is the covariance matrix between the $J$ observations calculated based on their pair-wise distance using the local variogram parameters (Eq. 3), and $\boldsymbol{c_0}$ is a vector for the covariance of the field at location $x_0$ and the various $x_i$. The total measurement noise and micro-scale variability for each observation is represented by the noise matrix $\boldsymbol{N}$. This is a diagonal matrix whose entries on the diagonal are the squared observation uncertainties, i.e., $\boldsymbol{N} = \boldsymbol{\tau}\boldsymbol{\tau}^T$, with $\boldsymbol{\tau}$ being the vector of observation uncertainties. To ensure that the weights sum to 1, an extra row and column is added to the Kriging matrix equation, where $\boldsymbol{w_0}$ now are the final Kriging weights, and $\varphi$ is the so-called Lagrange multiplier (Hengl, 2009):

$$\boldsymbol{\lambda_0'} = \begin{bmatrix} \boldsymbol{w_0} \\ -\varphi \end{bmatrix} = \begin{bmatrix} \boldsymbol{C} + \boldsymbol{N} & \boldsymbol{1} \\ \boldsymbol{1}^T & 0 \end{bmatrix}^{-1} \begin{bmatrix} \boldsymbol{c_0} \\ 1 \end{bmatrix}$$

We also estimate the prediction uncertainty of the interpolated elevation anomaly field. The uncertainty primarily depends on the distance to (and uncertainty of) the nearest data points, along with the covariance structure informing on how much the field is correlated at those distances. Two kinds of uncertainties may be computed: The uncertainty of the underlying field value, and the predictability of new data points obtained at the same location. The latter also accounts for the uncertainty associated with the micro-scale variability of the field and measurement uncertainties. The variance of with the interpolated elevation anomaly field at location $x_0$ can be estimated as the weighted average of covariances between $x_0$ and all observation points $x_i$. Additional uncertainty from the Lagrangian correction term is adjusted for by adding its value (Hengl, 2009):

$$\mathrm{var}[\hat{f}(x_o)] = \sigma^2 - \boldsymbol{c_0}^T \cdot \boldsymbol{w_0} + \varphi \tag{6}$$

with $\sigma^2$ being the local variance of the field, as determined from the local variogram. To estimate the variance of new observations $z_0$ obtained at the same location $x_0$, also the observation uncertainty must be addressed, and a nugget term is added to the equation (Paciorek, 2008):

$$\mathrm{var}[\hat{z}(x_o)] = \mathrm{var}[\hat{f}(x_o)] + \sigma_n^2 \tag{7}$$

This is the uncertainty to be used for validating the resulting DEM (section 7.1).

Applying the above equations using the previously determined spatially varying variogram parameters, we derive estimates for interpolated elevation anomalies (Eq. 5) and associated field variances (Eq. 6) at all PRODEM grid cell centres. To improve the estimation of the mean elevation anomaly across the grid cell, we subsequently smooth the elevation anomaly field by applying a 3x3 grid cell mean filter on the elevation anomalies obtained for grid cell centres. The resulting anomaly field and associated uncertainty is shown in Figure 7a-b. For each grid point, we also apply Eq. 4 to derive the weighted average acquisition time of observations (in units of day-of-year) used in the elevation prediction (Fig. 7c). This provides an assessment of the average day during the four-month summer period that the local interpolated elevation surface is most representative of, thereby providing a daily time stamp with each pixel in the DEM. This map of day-of-year values will be useful, for example, if generating elevation change maps by differencing individual PRODEMs.

Finally, the ArcticDEM 500 m mosaic is added to the obtained elevation anomaly field, forming an annual summer PRODEM for the marginal ice sheet zone (Fig. 7d). In a few places, the interpolated anomalies give rise to negative elevation values, which is unrealistic. This is caused by boundary effects; limited data leads to a retrieved constant dh-field in an area close to the margin where ArcticDEM displays large variability. For these few areas, we correct the PRODEMs accordingly.

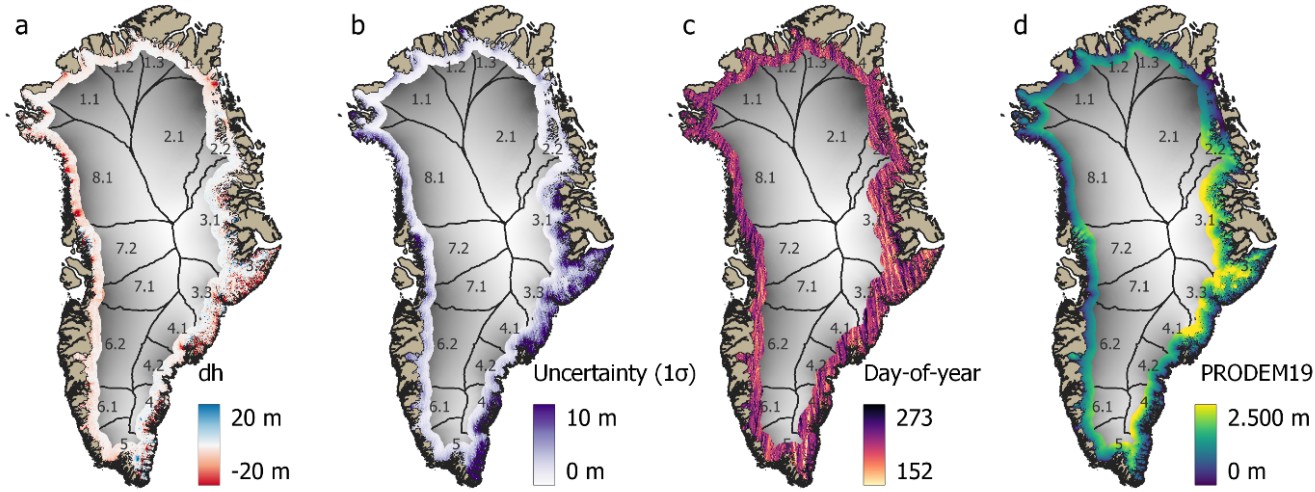


**Figure 7: PRODEM19 results. a) The resulting field elevation anomalies (dh) relative to ArcticDEM, along with b) the associated uncertainty (1σ) on the interpolated field, c) the weighted day-of-year of the interpolated surface, going from June 1st (day: 152) to September 30th (day: 273), and d) the resulting PRODEM19 elevations. ArcticDEM elevations (greyscale) are used for background, with drainage basins indicated.**

**6 General attributes of the PRODEM series (2019-22)**

Figure 8 shows the distributions of PRODEM19 elevation anomalies relative to ArcticDEM (Fig. 8a), the uncertainty of the interpolated elevation fields (Fig. 8b), and the associated time stamp of the derived surfaces (Fig. 8c). Distributions are shown for the entire PRODEM area as well as for each major drainage basin. Over the current four-year PRODEM series, the various distributions evolve, but the dominant patterns persist (Fig. 9).


The distribution of elevation anomalies (Fig. 8a) is distinctly leaning towards negative values (i.e., implying a general lowering of the surface relative to ArcticDEM) with a median value for the entire PRODEM area of -0.9 m (2019 values). The spread of the distribution of elevation anomalies is large, however, and with a heavy tail towards negative values indicating that large areas in the PRODEMs have considerably lower elevations than ArcticDEM. We attribute this to an actual surface lowering

of the ice sheet that has taken place throughout and subsequent to the acquisition period (2007-2022) of stereo imagery utilized for generating ArcticDEM. Spatially, negative anomalies (decreased elevations) dominate the West Greenland marginal areas (Fig. 8a; green curves), while regions with positive anomalies (increased elevations) are more common in East Greenland (Fig. 8a; blue curves), as it can also be seen from Figure 7a. For all drainage basins, however, the median elevation anomaly is negative.

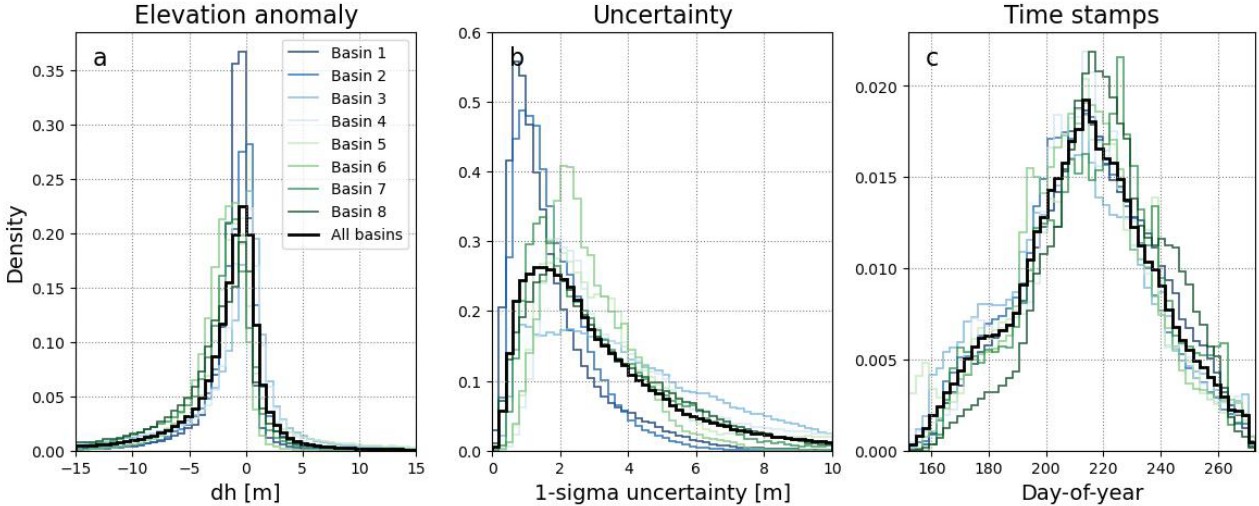


**Figure 8: Distribution of a) PRODEM19 elevation anomalies to ArcticDEM, b) uncertainties on the PRODEM19 elevation field, and c) weighted day-of-year of the altimetry used as input for the interpolation. Histograms are shown for the entire PRODEM area (black lines) as well as individually for all major drainage basins, with blue colours for drainage basins in North and East Greenland, and green colours for basins in South and West Greenland. Colour saturation is relative to basin latitude.**

The overall spatial patterns are stable over time. The median value of the elevation anomaly distribution for a drainage basin may change over time, often showing a decreasing trend (indicative of overall decreasing elevations), but the differences between values for individual drainage basins are significantly larger than the change over time within a single basin (Fig. 9a).

A map of the accumulated elevation change since 2019 can be calculated by comparing the spatial map of elevations (or

elevation anomalies) to the corresponding 2019 values ($\Delta h_{year;2019} = h_{year} - h_{2019} = dh_{year} - dh_{2019}$). The median of the $\Delta h_{year;2019}$ distribution across the entire PRODEM area shows a steady elevation decrease amounting to a total average decrease of 0.6 m over the three-year period from summer 2019 to summer 2022 (Fig. 9b). The southern and south-eastern drainage basins display the strongest decreasing trend, with elevations on average decreasing by 1.8 m (basin 5) and 1.3 m (basin 4). The northern and western regions (drainage basins 1, 6, 7, and 8) also tend to experience substantial decreases in

elevation of between 0.4 to 0.9 m. On the contrary, surface elevations in marginal North East Greenland (basins 2 and 3) show little total elevation change over the PRODEM period. This is primarily due to a substantial increase in surface elevations within this region from summer 2021 to 2022; the year during which the southern and south-eastern basins (basins 5 and 6) experience the strongest decrease in surface elevation. These general spatial trends are in line with those previously reported in the literature (e.g., (Jay Zwally et al., 2011; Sørensen et al., 2018)), but the PRODEMs allow to investigate the temporal

aspect of these changes in much higher detail that previously possible. We furthermore observe that the heterogeneity is high within each region, with areas of increasing and decreasing surface elevations existing side by side. To summarize; while surface elevations in the marginal areas of the Greenland Ice Sheet are generally decreasing since 2019 across almost all regions and all years, the magnitude of change show large spatial and annual variability, as the spatial pattern of change is complex and overlaid with substantial inter-annual variations.

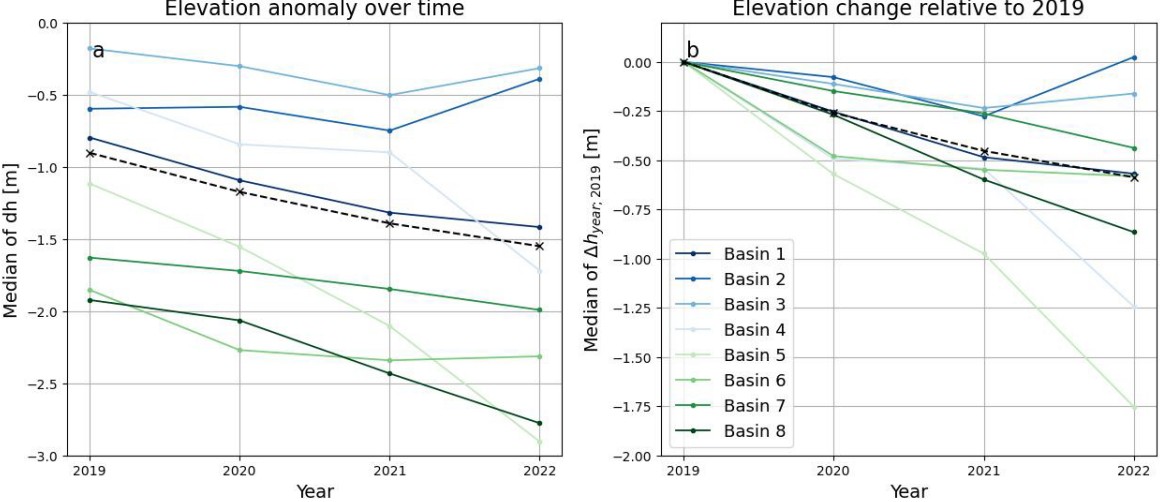

**Figure 9: a) Evolution in median elevation anomaly relative to ArcticDEM over the PRODEM period (2019-2022), and b) Evolution in median of the total elevation change relative to 2019 over the PRODEM period. Values are given both on drainage-basin scale (coloured from blue to green, with blue being marginal drainage basins in North and East Greenland, and green being basins along the South and West Greenland coast; darker colours correspond to higher latitudes), and for the entire PRODEM area (black dashed line).**

Areas with limited data availability result in increased uncertainty of the interpolated surface. The median $1\sigma$ uncertainty of the PRODEM19 elevation anomaly field (and hence also the PRODEM19 elevations) is 2.6 m, and with a mode of 1.4 m (Fig. 8b). Similar values are obtained for subsequent years. However, the distribution has an extended tail towards higher values, implying that some areas cannot be interpolated very accurately based on the available data. For all years, this is particularly an issue for drainage basin 3, for which an exceptionally long tail exists, corresponding to an extended area of relatively large uncertainties. Given denser data close to the pole due to the satellite orbits, the uncertainty generally decreases towards the north.

## 7 Validation of the PRODEMs

We assess the robustness of the PRODEMs through two complementary approaches: Block Leave-On-Out Cross-Validation (Block LOO-CV) and validation against an external data set. Block LOO-CV (Section 7.1) provides an internal assessment of the product performance by iteratively interpolating the elevation anomaly field based on all but data from a single track, and subsequently testing on the omitted sample. This method allows reliable validation with good spatial distribution across the entire PRODEM area, and it provides robust estimates of the product performance with respect to the input data. With its superior coverage, the distribution of prediction errors obtained from the Block LOO-CV further allows an in-depth analysis of the spatial error structure, shedding light on the product limitations and potential areas for improvement (Section 7.2).

Validation against an independent external data set offers an additional layer of verification, as it allows to identify potential biases or inconsistencies inherent in the original dataset. However, independent validation data from the PRODEM region and period is sparse. In section 7.3, we evaluate the PRODEM19 product performance against a single CryoSat Validation EXperiment (CryoVEX) flight-line of airborne laser scanner data acquired in August 2019 covering a short section across the Northeast Greenland Ice Sheet margin (Hvidegaard et al., 2021).

## 7.1 Block Leave-One-Out Cross-Validation

When validating the PRODEMs using Block LOO-CV, we iteratively remove a subset of observations, and the elevation anomaly at the location of the removed data points is predicted based on the reduced data set, thus allowing an evaluation of the prediction error. The subset of removed data points consists either of one track of CS2 data or IS2 data (one track consisting

of all three strong beams), as these are correlated. Due to computational limitations, we refrain from performing a full analysis, in which this process should be repeated on subsets consisting of all tracks independently, and we limit the number of repetitions to maximum 50 times for each basin, after which the results are found to stabilize. On average, one track of removed data corresponds to 0.5 % (CS2) and 1.8 % (IS2) of the total data set within a basin. The prediction errors (Fig. 10a) are

subsequently normalized relative to the predicted standard deviation of new observations obtained at the given locations (Eq. 7) combined with the uncertainty of the data used for validation (Section 4). If elevations are well estimated within their associated uncertainties, the normalized prediction errors should be distributed according to the standard normal distribution.

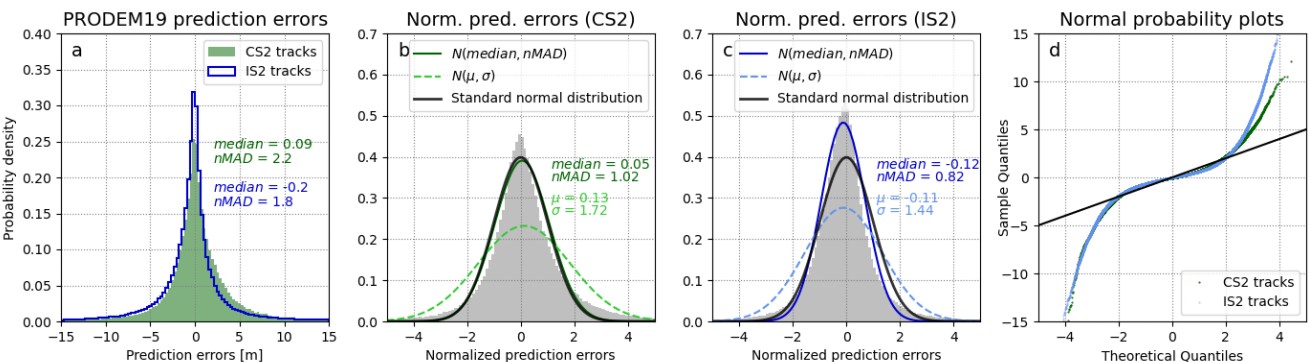

**Figure 10: Distribution of PRODEM19 prediction errors of the elevation anomaly after removal of either one track of CS2 data, or
one track (consisting of three beams) of IS2 data. a) Distribution of prediction errors prior to normalization. b, c) Distribution of
prediction errors when normalized relative to the estimated 1σ uncertainty of the interpolated field and the observation uncertainty
of the validation data. For both distributions are shown also the associated normal distributions based on calculating the median
and normalized MAD of the prediction errors (solid coloured lines) as well as using the mean and standard deviation (dashed
coloured lines). The standard normal distribution (solid black line) is shown for comparison. d) Normal probability plots comparing
the quantiles of the prediction error distributions with those of the standard normal distribution. Deviations from a straight line
(black) indicate departures from normality.**

Figure 10b-c shows the normalized prediction error distributions for PRODEM19 after successive removal with resampling of one track of CS2 and IS2 data, respectively, within each drainage basin. Prior to normalization, the prediction error distributions are strongly peaking around 0 m (Fig. 10a; Table 1): Median of the distributions are close to 0 m and with a

nMAD around 2 m. After normalization, however, both sets of normalized prediction errors are reasonably well described by the standard normal distribution, suggesting, as desired, that large prediction errors tend to be associated with large prediction uncertainties. Similar results are obtained for subsequent years in the PRODEM series (Table 2), and on average 63 % (CS2) and 72 % (IS2) of the predicted elevations fall within their associated 1σ uncertainty interval. The slightly lower percentage obtained for the CS2 prediction errors than the theoretically expected value (68 %) may partly be due to remaining uncertainties

in the CS2 validation data. With the prediction errors of the elevation fields well distributed according to the prediction uncertainty, we conclude that the PRODEMs within their associated uncertainties generally provide a good description of the true elevation fields.

| Distribution of prediction errors | | 2019 | 2020 | 2021 | 2022 |
|---|---|---|---|---|---|
| **CS2** | **median ($\mu$)** | 0.09 (0.32) | 0.07 (0.24) | 0.22 (0.39) | 0.10 (0.24) |
| | **nMAD ($\sigma$)** | 2.2 (6.1) | 2.1 (6.0) | 2.5 (6.9) | 2.2 (6.6) |
| **IS2** | **median ($\mu$)** | -0.20 (-0.12) | -0.16 (0.12) | -0.27 (-0.12) | -0.16 (0.13) |
| | **nMAD ($\sigma$)** | 1.8 (7.7) | 1.8 (7.9) | 1.8 (7.8) | 1.7 (7.7) |

**Table 1: Parameters describing the distribution of prediction errors after cross-validation based on successive removal with
replacement of one track of CS2 or IS2 data, respectively, for each year in the current PRODEM series.**

The analysis, however, also identifies some limitations with the reported uncertainties: All distributions display higher central peaks than that of a standard normal distribution and significantly heavier tails (Fig. 10d). The central clustering of prediction

errors implies that most elevation uncertainties tend to be slightly over-estimated. The heavier tails, on the other hand, indicate that the reported uncertainties do not capture the full range of variability, and, consequently, that errors in elevation estimate substantially larger than the prediction uncertainty are much more frequent than theoretically expected. A potential explanation for the existence of these heavy tails is that the equation used to derive the uncertainty field (Eq. 6) assumes that the uncertainty associated with the input observations conforms to a Gaussian distribution. This assumption of normality is, at best, an approximation. The observed elevation difference distributions near intersecting satellite tracks employed for assessing the observation uncertainty show a much higher central tendency than a normal distribution, particularly in areas of high roughness (Figure 2a,c). We hypothesise that the enhanced probability densities at the tails of the observation uncertainty distributions are carried over to the interpolated field.

The distribution of prediction errors, raw as well as normalized, display slight differences depending on which set of satellite observations are used for validation. We consider the prediction errors based on validation against the more accurate IS2 data to best reflect the elevation field uncertainties, and base the following analysis on these (Section 7.2). The IS2 normalized prediction errors (Fig. 10c) are slightly more centrally distributed, i.e., they tend to have errors smaller than the reported uncertainty. We observe that depending on which data set is used for validation, there is a tendency towards a slight overestimation (for CS2) or underestimation (for IS2) of the observed elevations. This feature is fairly robust across years and drainage basins, albeit with large regional differences in magnitude, and it is likely due to the construction of the PRODEMs from two sensors that may interact slightly differently with remaining snow/firn-covered surfaces within the PRODEM area.

| Distribution of normalized prediction errors | | 2019 | 2020 | 2021 | 2022 |
|---|---|---|---|---|---|
| CS2 | median ($\mu$) | 0.05 (0.13) | 0.04 (0.12) | 0.11 (0.20) | 0.06 (0.13) |
| | $nMAD$ ($\sigma$) | 1.0 (1.7) | 1.0 (1.7) | 1.1 (1.9) | 1.1 (1.8) |
| | Within standard central region ($\pm1$) | 64 % | 64 % | 61 % | 62 % |
| IS2 | median ($\mu$) | -0.12 (-0.11) | -0.10 (-0.07) | -0.16 (-0.15) | -0.11 (-0.07) |
| | $nMAD$ ($\sigma$) | 0.8 (1.4) | 0.8 (1.5) | 0.8 (1.4) | 0.8 (1.5) |
| | Within standard central region ($\pm1$) | 71 % | 72 % | 72 % | 71 % |

Table 2: Parameters describing the distribution of normalized prediction errors after cross-validation based on successive removal with replacement of one track of CS2 or IS2 data, respectively, for each year in the current PRODEM series. In the ideal case, the normalized prediction errors should form a standard normal distribution, e.g., median and mean value ($\mu$) equal to 0, $nMAD$ and standard deviation ($\sigma$) equal to 1, and with 68 % of the data falling within $\pm1$ of the distribution (central region with $\sigma = 1$). For both satellites and all years, the distribution of normalized prediction errors shows good consistency with these values.

### 7.2 Analysis of the PRODEM error structure

### 7.2.1 Predictive capability of PRODEMs depending on location and terrain properties

While the Block LOO-CV shows that the PRODEM elevation fields overall are well described within the prediction uncertainties, the product performance may vary spatially depending on the physical location and the terrain properties. We stratify the PRODEM IS2 prediction errors (raw as well as normalized) relative to the following parameters: Location (latitude/longitude), elevation, roughness, slope, aspect, uncertainty of ArcticDEM, and long-term trends in summer elevation (Fig. 11). Elevation, aspect, slope and ArcticDEM uncertainty are based on the 500-meter resolution ArcticDEM mosaic, roughness is based on the 100-meter resolution ArcticDEM mosaic, and long-term summer elevation trends are from Slater et al (Slater et al., 2021a). For most parameters, the errors are well characterized by the reported uncertainties. As an example, the dispersion of the prediction error distribution tends to decrease with increasing latitude (due to denser data coverage),

whereas the dispersion of normalized prediction errors stays relatively constant around unity (Fig. 11a). No significant relationship exists between prediction errors and longitude (not shown).


The largest prediction errors are found in areas of rough terrain (Fig. 11c): The prediction errors linearly increase with terrain roughness, attaining nMAD values of the error distribution of up to 10 m in areas of extreme roughness (ArcticDEM 100-meter scale roughness values of 30 m, roughly corresponding to the 95[th] percentile of the roughness field within the PRODEM area). This is not unexpected; At such high roughness it is difficult to properly interpolate an average 500 m elevation field,

and furthermore the data used for interpolation is often limited to IS2, since CS2 does not provide good measurements in such terrain. Similar relationship is found for the slope (not shown). The prediction errors also reach increasingly high values as the ArcticDEM uncertainty (measured as the MAD of contributing DEMs) increases (Fig. 11e). As the ArcticDEM mosaic is used as a reference during PRODEM construction, it is not surprising that these uncertainties will propagate to the PRODEMs. In order of decreasing importance, the prediction errors also tend to be higher in areas of low elevation (Fig. 11b), for areas

displaying large negative trends in elevation over the summer (Fig. 11f), and for south-facing slopes (i.e. aspect ~180°; Fig. 11d).

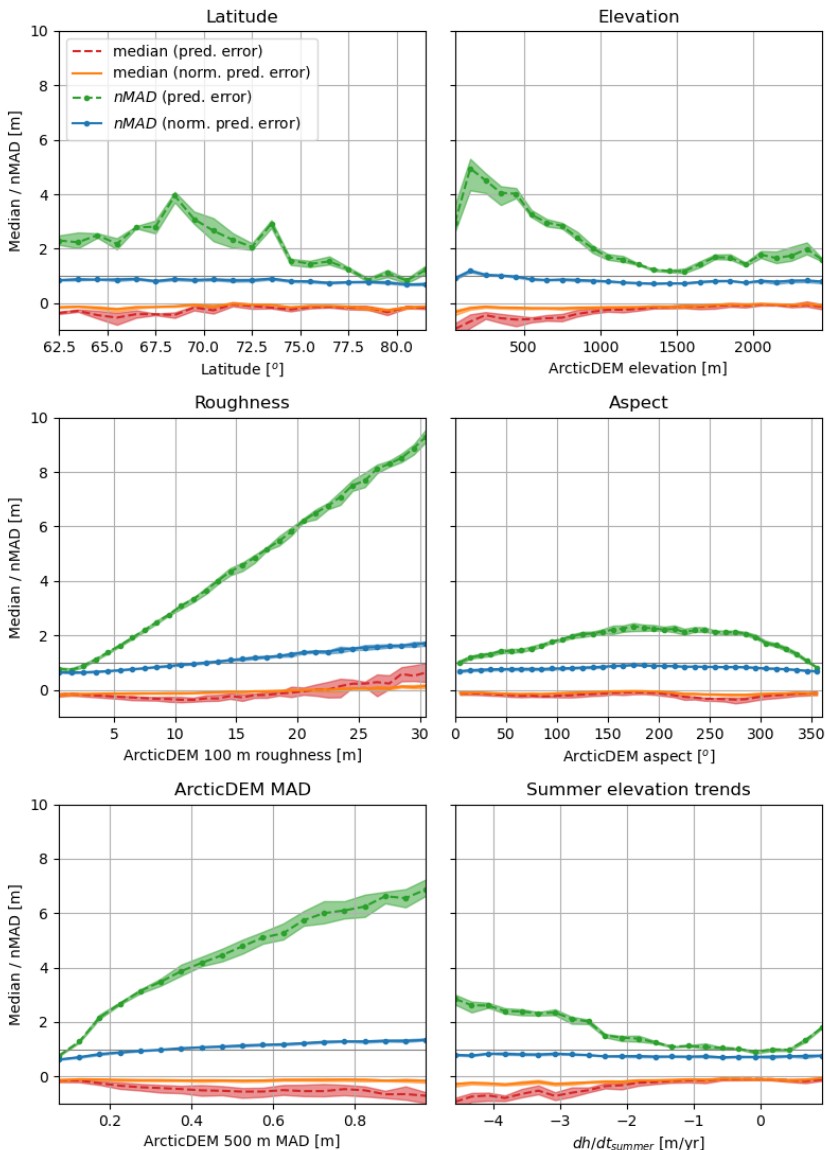

**Figure 11: Descriptive measures (median and nMAD) of the distribution of PRODEM19 raw (dashed lines) and normalized prediction errors (solid lines), when stratified relative to various location and terrain properties that may impact the PRODEM**
**product performance. Lines correspond to mean values over the four-year PRODEM series, with the envelope showing minimum and maximum values for the four years. Bins are created to cover the 5 to 95 percentiles of each data set. The ArcticDEM MAD value is computed as the median absolute deviation of the DEMs contributing to ArcticDEM at a given location, and therefore provides an uncertainty estimate of ArcticDEM. Summer elevation trends from Slater et al** (Slater et al., 2021a)**.**

The analysis allows us to identify problematic areas, where one should be cautious about the reported PRODEM uncertainties.

These are characterized by having nMAD values of the normalized prediction uncertainties significantly above one, which we here define as larger than 1.2. This is the case for areas with roughness above 17 m and/or ArcticDEM MAD values above 0.65 m. The first condition covers 18 % of the PRODEM area, while the second condition covers 13 %, and they largely overlap in the areas along the edge of the ice sheet (in 24 % of the PRODEM area, at least one of the conditions is met).

### 7.2.2 Spatial correlation of errors

The errors of the PRODEM elevation fields are not randomly distributed. In addition to the influence of location and varying terrain properties (Section 7.1.1), we observe a spatial correlation in the error structure. A main reason for this is that the observations are acquired at different times during the summer, during which the surface elevation may change non-negligibly due to surface mass balance processes. Consequently, areas with closely spaced IS2 tracks acquired early and late in the season may result in stripes in the resulting DEM (see Discussion).


To gain an understanding of the spatial structure of the elevation prediction field, we analyse the spatial correlation of the IS2 prediction errors. Since the error structure is likely to be aligned with the satellite tracks (in particular the straight lines of IS2 observations), we construct directional variograms along and across the IS2 track directions. We again compute the empirical variograms using the Dowd estimator (Mälicke et al., 2022). As the track direction is location dependent, the analysis is done

for each basin individually (both for ascending and descending tracks), and the basin-scale variograms are averaged by taking the median of the semi-variances before fitting an exponential variogram without nugget to the results. In the along-track direction, the error structure has an effective correlation length of about 3 km (length scale: 1000 m), whereas the variogram for errors in the across-track direction displays a small length scale (335 m), indicating that errors are essentially uncorrelated in this direction (Figure 12).

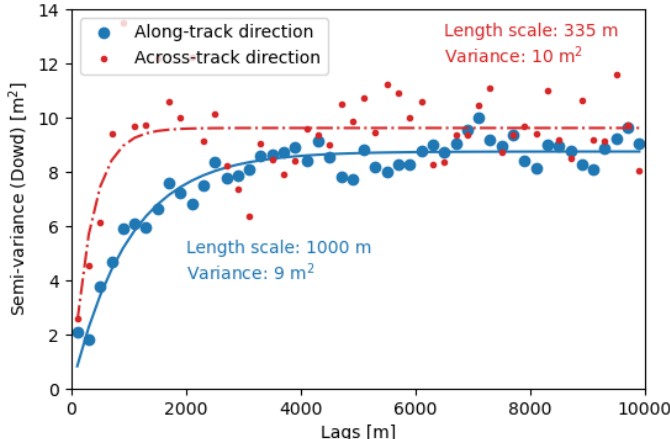


**Figure 12. Directional variogram showing the spatial correlation structure of IS2 block LOO-CV prediction errors in the along-track and across-track directions. In the across-track direction, the variogram has a small length scale, indicating that in this direction errors are essentially uncorrelated. Based on data from 2019.**

### 7.3 Validation of PRODEM19 against airborne laser-scanner measurements

Independent validation data that may be used for validation of the PRODEMs is sparse since airborne altimetry campaigns usually take place in Spring, i.e. outside the PRODEM period. In 2019, however, the ESA CryoVEx/ICESat-2 spring campaign suffered from bad weather, and, as a result, an airborne summer measurement campaign with a laser scanner set-up was conducted in August 2019 (Hvidegaard et al., 2021). The focus of this campaign was sea ice in the Wandel Sea, northeast of Greenland, but one flight line captured on August 10[th] (ALS_20190810T151333_164135 (European Space Agency, 2022))

also covers a section of land ice, thus providing independent Airborne Laser Scanner (ALS) data to be used for validating PRODEM19.

We first down sample by averaging the very high resolution CryoVEX ALS data to 10 m grid resolution and filter it to only contain ice surface observations. Subsequently, we average all ALS ice elevations within each PRODEM grid cell and compute

the difference to the gridded PRODEM19 elevation value. Statistics for the resulting distribution of elevation differences ($\Delta h$) are provided in Table 3. While this is the most direct way to compute the elevation differences, it does, however, not account for uncertainties related to the number and distribution of ALS data within a grid cell. We therefore also compare against a reduced ALS data set constructed with additional scrutiny regarding the number of observations used to produce the gridded ALS elevations: We remove ALS average elevations in the 10-meter resolution grid based on less than 5 observations, and

after subsequent down sampling to the PRODEM 500-meter resolution grid, averages based on less than 50 contributing elevation measurements are removed. This causes a slight reduction of the available grid cell values to be used for comparison, but it removes a significant number of outliers, leading to less dispersion of the elevation difference distribution. Another way to account for a potential uneven distribution of ALS observations across the PRODEM grid cells is by computing the elevation differences based on anomalies to ArcticDEM ($\Delta dh$), thereby adjusting for the local slope: A 10-meter gridded set of ALS

elevation anomalies is computed, followed by down sampling to the PRODEM grid. The ALS elevation anomalies are subsequently compared to the PRODEM elevation anomalies. This approach gives rise to substantially less dispersion of the observed elevation difference distribution.

Regardless of the approach, the differences between ALS and PRODEM19 elevations are small (Table 3). The statistics

slightly improve when removing gridded ALS data based on limited observations, and when conducting the analysis on the difference in elevation anomalies rather than the elevations directly.

| | Mean [m] | Median [m] | nMAD [m] | SD [m] | RMSE [m] | Number of grid cells |
|---|---|---|---|---|---|---|
| $\Delta h = PRODEM19 - ALS$ | 0.15 | 0.16 | 2.0 | 4.1 | 4.1 | 1283 |
| $\Delta h_{red} = PRODEM19_{red} - ALS_{red}$ | 0.15 | 0.17 | 1.7 | 2.9 | 2.9 | 1141 |
| $\Delta dh = dh_{PRODEM} - dh_{ALS}$ | 0.14 | 0.15 | 1.0 | 2.2 | 2.2 | 1283 |
| $\Delta h_{norm} = \dfrac{\Delta h}{\sigma_{PRODEM}}$ | 0.13 | 0.12 | 1.1 | 2.5 | 2.6 | 1283 |
| $\Delta h_{red,norm} = \dfrac{\Delta h_{red}}{\sigma_{PRODEM}}$ | 0.13 | 0.12 | 1.0 | 2.0 | 2.0 | 1141 |
| $\Delta dh_{norm} = \dfrac{\Delta dh}{\sigma_{PRODEM}}$ | 0.12 | 0.11 | 0.6 | 1.3 | 1.3 | 1283 |

**Table 3: Statistics for comparison of elevations (raw and normalized) in PRODEM19 and CryoVEX ALS data from Northeast Greenland acquired on August 10[th], 2019. The ALS data is first down sampled to the PRODEM grid. Standard deviation (SD) and**

**root-mean-square error (RMSE) are calculated as: $SD = \sqrt{\frac{\sum(\Delta h - mean)^2}{N-1}}$ and $RMSE = \sqrt{\frac{\sum \Delta h^2}{N-1}}$, where N is the number of cells.**

The ALS flight line crosses areas of varying roughness (Fig. 13), including areas of high topographic relief associated with substantial PRODEM elevation uncertainties. Using a similar approach as for the Block LOO-CV prediction errors (section 7.1), we therefore also investigate the distribution of normalized errors, i.e. the elevation differences divided by the local PRODEM elevation field uncertainty. Also here, the resulting distributions are close to a standard normal distribution (Table

3), implying that PRODEM19 provides a good estimate of the elevation field within the stated uncertainties.

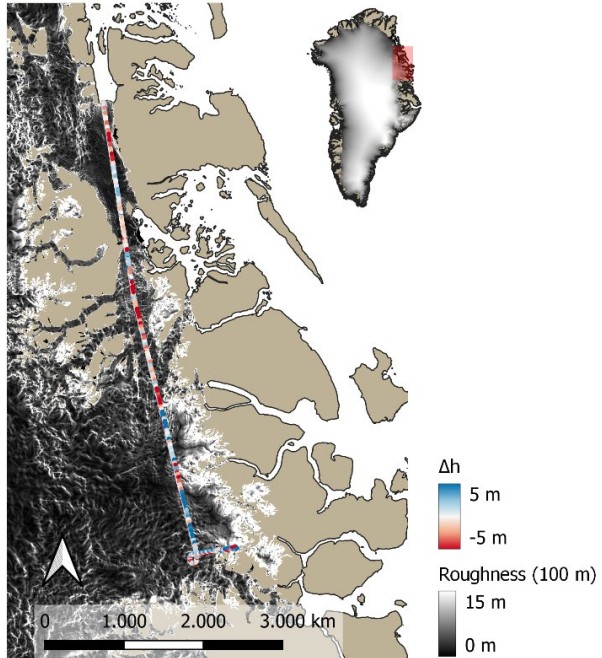

**Figure 13: Location of the CryoVEX ALS flightline from August 2019 and the gridded ice surface elevation differences of PRODEM19 to the ALS data. ArcticDEM 100-meter scale roughness is used as background (greyscale).**

**8 Comparison to an IS2-only Greenland DEM obtained for the period 2018-19**

Very few large-scale DEMs exist for the Greenland Ice Sheet that is valid for the PRODEM period. The exception is a recently-constructed DEM derived from IS2 data from November 2018 to November 2019 (in the following abbreviated IS2DEM19), developed based on a spatiotemporal fitting model (Fan et al., 2022). For several reasons we do not expect the IS2DEM19 to be identical to PRODEM19. A significant difference is that IS2DEM is based on data from an entire year, during which the surface elevations evolve with the seasons. A larger snowpack during the winter season is expected to result in generally higher

elevations in IS2DEM19 than in PRODEM19. This is corroborated by a comparison of the two elevation fields; their median elevation difference across the marginal ice sheet zone is -2.0 m, with the negative sign indicating that PRODEM19 tends to display lower elevations. However, substantial variability in the elevation differences is observed across the area (Fig. 14a).

Further, the PRODEMs are derived as elevation anomalies to ArcticDEM, whereas such an approach has not been taken for

IS2DEM19. Looking at the small-scale features of the DEMs, this difference in approach becomes evident, with many of the fine-scale features existing in ArcticDEM being preserved in the PRODEM elevations (albeit in a modified form). As a result of its smoother appearance, the slope distribution of IS2DEM19 tends towards smaller values than PRODEM19 and ArcticDEM, the latter two having almost identical slope distributions (Fig. 14h).

Figure 14 shows how the large- and fine-scale structures differ between the three DEMs (PRODEM19, IS2DEM19, and ArcticDEM v4.1) for Petermann Gletsjer in Northern Greenland. Their differences are visible directly from the elevation field (e-g), but they become even more evident from the spatial patterns of elevation differences to PRODEM19 (b-d) and slope (i-k). The fine-scale structure of PRODEM19 is fairly similar to that of ArcticDEM, whereas its large-scale field is more similar to IS2DEM19. This is especially evident for an area in the outer part of the glacier tongue, where both PRODEM19 and

IS2DEM19 display substantially higher elevations than ArcticDEM, suggesting that the glacier front has advanced since acquisition of the ArcticDEM elevation field. The resemblance between the PRODEM19 and IS2DEM19 elevation models is improved by first smoothing PRODEM19 with a Gaussian filter (Fig. 14b), after which the median difference between the two elevation fields slightly decreases (median difference for the marginal ice sheet zone: -1.9 m). Compared to IS2DEM19, the

increased resolution of fine-scale features in the PRODEMs comes at the cost, however, of additional assumptions regarding
the stability over time of the small-scale structure of the elevation field.

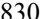

**Figure 14: A comparison of the three elevation fields: IS2DEM19, PRODEM19 (including a filtered version; PRODEM19$_{filt}$), and ArcticDEM across Petermann Gletsjer. PRODEM19$_{filt}$ is produced by applying a Gaussian filter with 5 pixels radius and relative standard deviation of 50 %. Coverage of the three DEMs differs slightly across the region. a) A histogram of the distribution of elevation differences across the PRODEM area. b-d) Maps showing the spatial distribution of elevation differences of PRODEM19 versus IS2DEM19 and ArcticDEM, respectively, across Petermann Gletsjer, along with a comparison of elevation differences between PRODEM19$_{filt}$ and IS2DEM9. e-g) The elevation fields across Petermann Gletsjer. h) Distribution of slopes across the entire PRODEM area, along with i) a table of mean and median values of the slope distributions. j-l) The spatial distribution of slope for the three elevation models across Petermann Gletsjer.**

## 9 Discussion

### 9.1 Impact of data sparsity on PRODEM elevation accuracy

For reliable outcome of Kriging, the field to be interpolated must be quasi-stationary, and hence it is a prerequisite to use a reference DEM to produce elevation anomalies for interpolation. Our approach has the advantage of inheriting the detailed elevation from ArcticDEM to maintain the high spatial resolution of the PRODEMs also in areas poorly covered by the satellite altimetry. Regardless of data coverage, the employed approach acts to add small-scale topographic features visible in ArcticDEM, with the underlying assumption that these are stable over time.

Ideally, the altimetry data should exhibit spatial variability representative of the elevation anomaly field, capturing both local fluctuations and broader trends. This will ensure that the Kriging interpolation for every grid cell is based on a sufficiently large set of representative samples. Based on the spatial correlation structure of the elevation anomalies, we can evaluate the number of observations that significantly contribute to the interpolated PRODEM elevation fields, i.e., observations located within a radius corresponding to the effective length scale of the local variogram. On average, the interpolated field is based on 78 nearby observations (median of the distribution; 2019 values) (Fig. 15). In 2019, a total of 4 % of the PRODEM grid cells suffer from sparsity in neighbouring data, which we take to be less than 5 observations within the local effective length scale. For 2 % of the grid cells, no observations exist within this interpolation radius. Slightly improved values are obtained for the subsequent years; PRODEM22 has the most well-determined elevation field, with less than 2 % of the grid cells suffering from sparse data (<1 % without any nearby observations). The issue of data sparsity is most pronounced in the southern regions; for the southernmost drainage basin (basin 5.0) the percentage with sparse data coverage has increased to 14 % (7 % without any nearby observations) (2019 values). In regions without any nearby observations, the ArcticDEM elevations are simply adjusted with a constant depending on the median value of the local anomaly field, as calculated from the 200 closest altimetry observations. Consequently, the PRODEMs are in these areas heavily reliant on the quality of ArcticDEM.

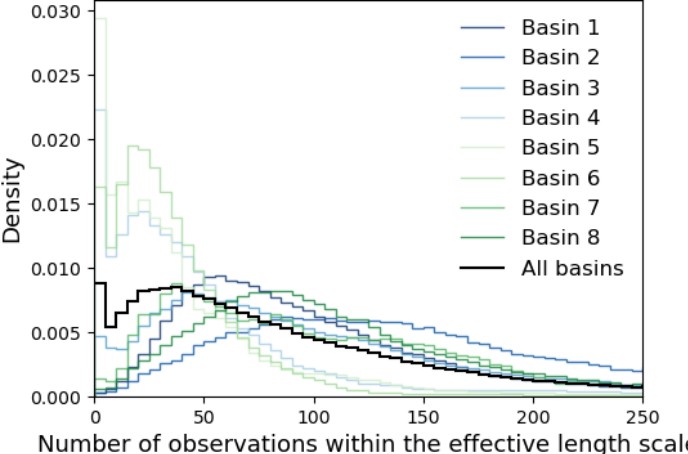

Figure 15: **Distribution of the number of altimetry observations within the interpolation radius for the PRODEM grid cells. The interpolation radius corresponds to the local effective length scale (roughly equivalent to three times the length scale for the exponential variogram model) of the spatial correlation structure of the elevation anomaly field. Histograms are provided for the individual drainage basins (coloured from blue to green, with blue being marginal drainage basins in North and East Greenland, and green being basins along the South and West Greenland coast; darker colours correspond to higher latitudes), and for the entire PRODEM area (black line). Data from 2019, which is the year when data sparsity is most pronounced.**

### 9.2. The PRODEM anomaly fields reveal artefacts in ArcticDEM

ArcticDEM is a multi-year elevation model that incorporates stereo photogrammetry from an extended period, during which the ice sheet topography has evolved. This leads to artefacts in the product, which become apparent in the interpolated elevation anomaly fields. Due to ArcticDEM being constructed as a mosaic of 100x100 km tiles, with each tile separately registered to

reference elevations from GrIMP DEM v2, the PRODEM elevation anomalies display a checkerboard pattern with each tile having a slightly different mean value. This is, for example, the case for an area in Central West Greenland (Fig. 16), where the distribution of anomalies in the northern tile (Fig. 16, tile a) is -7.5 m ± 7.9 m, whereas the distribution in the adjacent southern tile (Fig 16, tile b) is -0.3 m ± 9.7 m (PRODEM19 values). As apparent from this artificial checkerboard pattern, the derived field of elevation anomalies to ArcticDEM does not represent varying rates of elevation change rates.

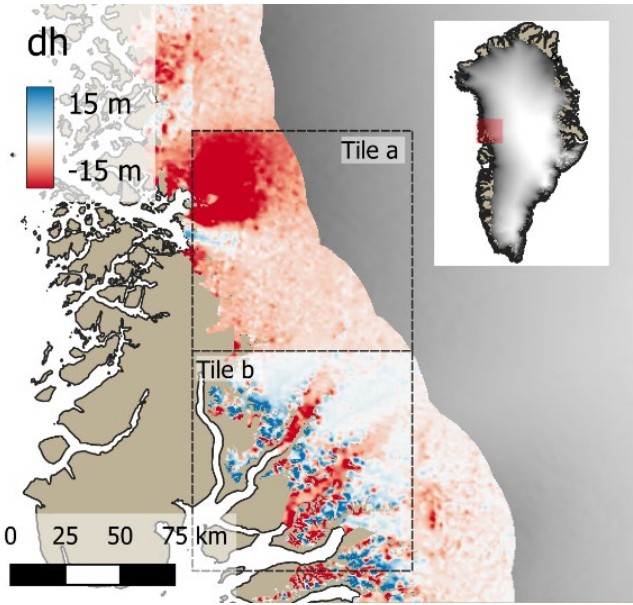


**Figure 16: The PRODEM elevation anomalies (here for PRODEM19) relative to ArcticDEM display a checkerboard pattern due to the way ArcticDEM was constructed. Two ArcticDEM tiles of 100x100 km are indicated.**

**9.3. Striped patterns in PRODEMs due to seasonal elevation changes**

Like ArcticDEM, the PRODEMs display artefacts caused by temporal differences in data acquisition, albeit on significantly
shorter time scales, resulting in less time for surface changes to occur. The PRODEM altimetry is collected during a four-month period, during which the surface elevation may change due to e.g., snowfall and surface melt. As a result, we in some areas observe a striped pattern in the interpolated PRODEM elevation anomaly field, seemingly aligned with the IS2 satellite tracks (Fig. 17a-b). The differences in the elevation anomaly field across the stripes are on the order of half a meter or less, and within the uncertainty of the interpolation.


Investigating the source of this pattern, we observe that it often occurs where two closely spaced IS2 satellite tracks are obtained at the beginning and end of the summer season, respectively. Indeed, the IS2 observations along the two tracks may differ in average elevation by up to a meter. For the area between the satellite tracks, the interpolated surface depends on which line of observations is closest. This can also be seen from the varying weighted mean acquisition time of the input data used in the
interpolation across the area: A striped pattern similar to that in the elevation anomalies is evident from the day-of-year field (Fig 17c). However, not all locations with large variability in average time of data acquisition are prone to form this stripy pattern. The pattern is only visible in areas where the surface elevation has changed significantly during the summer period, and where the spatial correlation of the elevation anomaly field is characterized by a short variogram length scale, so that primarily the closest data points are given a high weight during interpolation.


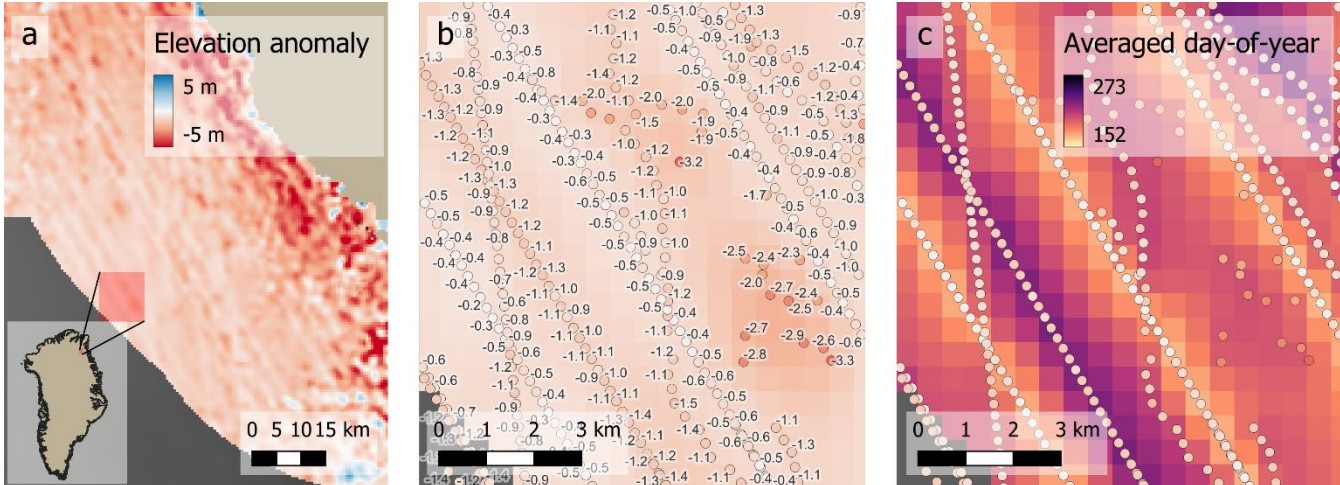

**Figure 17: A striped pattern in the PRODEM elevation anomaly fields is caused by differences in time of data acquisition. a) 2019 elevation anomalies in a small area of northeast Greenland (red box in the overview figure). b) A zoom-in (red box in a) of the elevation anomaly field to an even smaller area. Dots indicate the altimetry based on which the interpolation is based. The data is labelled and coloured according to the measured elevation anomaly (same colour scale as the interpolated field). c) Averaged day-of-year represented by the interpolated surface; going from June 1st (day: 152) to September 30th (day: 273) in 2019. The coloured dots represent the measured elevation anomalies.**

## 10 Data products and availability

The PRODEMs described in this manuscript can be accessed at https://doi.org/10.22008/FK2/52WWHG (Winstrup, 2023), with the manuscript based on the second release (v2) of the data set. Presently, the product covers summers 2019 through 2022. The product is expected to be updated annually over the coming years.

Each annual 500-meter resolution PRODEM is provided as a georeferenced raster in GeoTIFF-format (.tif). The rasters are in the National Snow and Ice Data Center (NSIDC) Sea Ice Polar Stereographic North projection and referenced to the WGS84 datum (EPSG:3413). Each PRODEM contains the following four data layers:

- *DEM:* PRODEMyy (heights above the WGS84 reference ellipsoid, in m)
- *variance:* Elevation uncertainty (field variance, in m$^2$)
- *dh:* Elevation anomalies (m) relative to the ArcticDEM v4.1 500 m mosaic
- *time stamp:* Time-stamp associated with the interpolation (in units of day-of-year).

## 11 Conclusion

We have constructed an annual series of 500-meter resolution summer DEMs (PRODEMyy) for the marginal zone of the Greenland Ice Sheet. The DEMs are created by fusing satellite altimetry data from CryoSat-2 and ICESat-2, and hence the PRODEM series starts in 2019, the first summer after ICESat-2 became operational. ArcticDEM v4.1 is employed as reference DEM, which ensures that high spatial resolution is maintained also in areas with limited altimetry. The present PRODEM series consists of four DEMs, with the most current being from summer 2022, and we aim to repeat the procedure for the coming years to continually obtain annual changes in surface elevation across this rapidly changing region of the Greenland Ice Sheet.

The PRODEMs are interpolated using regionally varying Kriging on elevation anomalies relative to ArcticDEM. Spatially-varying uncertainties of the individual observations are modelled based on analysing observed elevation differences close to intersecting satellite tracks. The applied approach is able to account for the large regional differences in observational uncertainty and covariance structure of the anomaly field across the area. The PRODEMs are validated using Block Leave-

One-Out Cross-Validation, and the obtained elevation fields are found to be well determined within their associated spatially-varying uncertainties. In most cases, the uncertainties are slightly conservative, but there is a higher number of outliers than predicted by a Gaussian distribution. The PRODEMs are best determined in the northern regions, where the data coverage is most dense, and lesser so in areas with very high roughness. PRODEM19 is additionally validated against data from one flight-line with an airborne laser scanner, and we find very good agreement between the two data sets.

For most of the marginal zone, we observe a lowering of the ice sheet surface compared to ArcticDEM, albeit with large spatial differences. However, since ArcticDEM is built from data from a regionally varying time window, the elevation differences between the PRODEMs and ArcticDEM cannot be directly converted to elevation change estimates. Indeed, some features in the elevation anomaly field merely reflect artefacts related to the construction of ArcticDEM in tiles: We observe a checkerboard pattern in the derived elevation anomalies, with each tile having distinctly different mean anomaly values.

Over recent years, the surface of the Greenland Ice Sheet has been experiencing a general lowering due to climate change, and on this trend is superimposed a complex and annually shifting pattern of changes due to glacier dynamics and mass balance processes. In almost all regions of the marginal ice sheet, we observe a thinning of the ice sheet. Most so in the south and south-eastern marginal areas of the Greenland Ice Sheet, where the average surface elevation has decreased by more than 1 meter over the three years. The exception is the marginal areas of North East Greenland, which from summer 2021 to 2022 experienced a substantial increase in elevation, causing the average surface elevation to almost be stable over the PRODEM period. By capturing the patterns of change in the marginal ice topography on an annual basis, the PRODEMs will give valuable insights into the inter-annual variability of ice sheet dynamic processes and contribute to the validation of ice sheet models.

**Author contributions.** MW developed the method, generated the DEMs, interpreted the results and wrote the manuscript with contributions from all co-authors. LSS initially conceived of the study and contributed with regular discussions and project guidance, the latter in collaboration with RSF. HR and SBS extracted datafiles with the IS2 and CS2 data and assisted with manuscript preparation. SHL, KDM provided valuable insights and significant revisions to the manuscript. All authors contributed to improving the final manuscript.

**Competing interests.** Ken Mankoff is a Chief Editor at ESSD. The authors declare that they otherwise have no conflict of interest.

**Acknowledgements.** ArcticDEM is provided by the Polar Geospatial Center under NSF-OPP awards 1043681, 1559691, and 1542736.

**Financial support.** This research has been supported by the Programme for Monitoring of the Greenland Ice Sheet (PROMICE).

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
