# Peer review of "PRODEM: An annual series of summer DEMs (2019 through 2022) of the marginal areas of the Greenland Ice Sheet"

_Earth System Science Data, 2023_

## Author Comment (AC1)

**Replies to reviewer comments by Romain Hugonnet**

We thank the reviewer very much for their appraisal of our manuscript and DEM product, which we truly appreciate.

We also appreciate the very constructive suggestions for improvements, and with these in mind we have clarified and adjusted the procedure as described below. The reviewer comments are shown below in blue, and our replies shown in black. Line numbers refer to the original version of the manuscript that has been reviewed.

Since the manuscript was initially submitted, several of the input datasets have been updated, and we have now included these:

- We use CryoSat-2 Baseline E for all years, as it is now available for the entire period
- ICESat-2 ATL06 has been updated to version 6; the new version has corrected the geolocation error found to exist in version 5
- ArcticDEM mosaic has been updated to v4.1 (as also proposed by the reviewer)

We have produced a new version of the PRODEMs using the updated data sets, and the text and figures in the manuscript have been adjusted accordingly. This includes e.g., an updated version of figure 10 now displaying another area that shows a similar checker-board pattern in elevation anomalies relative to ArcticDEM v4.1 (albeit less pronounced than previously found when comparing to ArcticDEM v3), and removal of figure 11 (including the related discussion) which is no longer relevant.

We hope that with these and the below changes, you will accept to review an updated version of the paper.

**Statement of expertise**

I have expertise in the aspects of this paper associated with:

- Remote sensing of surface elevation data for glaciology,
- Laser altimetry with ICESat-2,
- Geostatistical methods used for data fusion, in particular variography and kriging,
- Uncertainty analysis in a spatiotemporal context for the predicted elevations.

Would be nice to complement with another reviewer being specifically expert in:

- Radar altimetry with CryoSat-2 (though I do have some).

**General comment**

In their very well-written manuscript, the authors present a rigorously derived dataset of annual (2019-ongoing) gridded surface elevation at 500 m resolution for the Greenland Ice Sheet, predicted from spatiotemporally sparse ICESat-2 and CryoSat-2 elevations.

The study is a truly a great piece of work, and I commend the authors on their attention to detail at all steps of their analysis to produce a robust product for the community. My core expertise is mathematics and DEMs, in particular spatial statistics and uncertainty analysis, and this study is a rare sight in our field in that aspect, and I think the way forward so that other scientists and users can know how reliable a data product really is.

I still have several comments for the authors, which I think could improve some parts of their analysis substantially. Some might be a bit of work, but are important and should be OK given the author's knowledge in geostatistics, and others are optional.

Sorry if I cite my own work quite a bit in this review, there are not that many other studies looking at these specific problems in detail (geostats + uncertainty quantification + DEMs), and I hope it will be helpful to the authors!

**Main comment 1: Refine the modelling of heteroscedasticity of the predicted elevation**

Or "variability in error". This is something the authors have addressed quite a bit in the text and their estimates, in particular in Section 4.1 by constraining the variability in elevation error (CS2 - CS2) with surface roughness (for CS2; Figure 2b), and using the variability of 20 m IS2 points in the 250 m area.

However, Figure 8 shows that this modelling of the variability in error (including subsequent propagation with kriging) is not completely satisfactory:

- Too large predicted errors for the center of distribution,
- Yet does not capture the full range of variability (tails are quite significantly larger than a normal distribution).

Here, I recommend the authors several pists that should improve their reported error.

As the reviewer correctly points out, the uncertainty of the resulting elevation estimates showed large spatial variability (high heteroscedasticity) as expected due to varying data density and spatially-changing terrain. Despite our efforts to account for this, the results of our validation (Fig. 8) did indeed show that our kriging uncertainties did not fully capture the true variability of the deviations: In most areas, the uncertainties were slightly conservatively estimated, but a significantly higher number of outliers than expected existed.

In line with the changes suggested by the reviewer to improve this (see below), we have therefore meticulously adjusted the uncertainty calculation of the CS2 and IS2 measurements. We have further changed the allowed variogram nugget (also suggested by the reviewer; more about this later), which also influences the spatial structure of derived uncertainties.

We also now normalize the prediction errors relative to the total uncertainty of the prediction and the estimated observation uncertainty for the validation data. This causes the normalized error distribution to slightly improve – in particularly for the CS2 data, which are associated with larger observation uncertainties.

Despite all these changes, our cross-validation shows that the resulting error distributions from the PRODEM validation are very similar to previously. A figure showing a comparison of the associated Q-Q plots for the LOOCV differences is provided below, along with an updated version of Figure 8 (now Fig 9). The figure comparing Q-Q plots for the previous and revised PRODEM versions is only for illustrating the differences to the reviewer, and it is not included in the revised paper.

[Figure]

[Figure]

*Figure 9: Distribution of PRODEM19 prediction errors of the elevation anomaly after removal of either one track of CS2 data, or one track (consisting of three beams) of IS2 data. a) Distribution of prediction errors prior to normalization. b, c) Distribution of prediction errors when normalized relative to the estimated 1σ uncertainty of the interpolated field and the observation uncertainty of the validation data. For both distributions are shown the associated normal distributions based on calculating the median and normalized MAD of the prediction errors (solid lines) as well as using the directly calculated mean and standard deviation (dashed lines). The standard normal distribution (solid black line) is shown for comparison. d) Normal probability plots comparing the quantiles of the prediction error distributions with those of the standard normal distribution. Deviations from a straight line (black) indicate departures from normality.*

In other words, despite the enhancements in uncertainty estimation, some limitations persist within the reported error estimates. In the manuscript, L526-531 has been rewritten, so that this is clearly stated to ensure that the reader is aware of this issue. The section now reads as follows:

*The analysis, however, also identifies some limitations with the reported uncertainties: All distributions display higher central peaks than that of a standard normal distribution and significantly heavier tails (higher kurtosis). The central clustering of prediction errors implies that most elevation uncertainties tend to be slightly over-estimated. The heavier tails, on the other hand, indicates that the reported uncertainties do not capture the full range of variability, and, consequently, that errors in elevation estimate substantially larger than the prediction uncertainty are much more likely than theoretically expected.*

First, some small but important changes to avoid over-estimating the center distribution:

1. Consider using a Q-Q plot in Figure 8 (or add one), which will represent the tail differences in a clearer fashion,
   Q-Q plots have been added to Figure 8 (now Figure 9); see the updated figure above.

2. Avoid over-estimating the CS2 measurement error by dividing by square-root of 2 as the error source is combined twice in the cross-validation exercise (e.g., Equation 8, Hugonnet et al. (2022)).
   This has been corrected, and the following sentence has been added to the manuscript (L192):
   *As the distribution of elevation differences integrates the error sources from both elevation measurements, we derive an approximate measure of the 1σ uncertainty of the CS2 observations as:*
   $$\sigma_{spatial}^{CS2} = \frac{s_{MAD}}{\sqrt{2}} = \frac{1.4826}{\sqrt{2}} * MAD$$

3. Avoid over-estimating the IS2 measurement by taking a simple spread: separate segments will have largely uncorrelated errors, so you need to compute a standard error, i.e. divide by sqrt(N) with N the non-overlapping ICESat-2 segments.
   We have revised the approach used for computing the uncertainties associated with the IS2 measurements, so that it is now aligned with the method utilized for the CS2 data (more details below). Notably, we no longer rely on the spread of IS2 measurement along the 250m segments as an estimate for the terrain roughness, but instead use the terrain roughness calculated based

on ArcticDEM. In this respect, it is worth mentioning that the two roughness measures are highly correlated (correlation coefficient=0.8).

Then, to improve their modelled variability in error:

4. Have a consistent spatial variability dependency with terrain roughness for both IS2 and CS2 as it should be universal (e.g., Hugonnet et al. (2022) for slope + curv), you could apply the same scheme to both CS2 and IS2 (it sounds you did something similar for IS2 on L260-261? These statements are a bit unclear, consider explaining in more details, or adding a Figure if it tells something about the error!).

Inspired by the comments by the reviewer (above and below), we have made the following changes to the uncertainty calculations:

1) We now separate the uncertainty component into its spatial and temporal components
2) Elevation differences are now computed as the difference of elevation anomalies (instead of elevations); this approach accounts for a potential slope in the area.
3) We reduce the allowed spatial distance between measurements to 50m (instead of 100m)
4) The spatial component is computed only based on elevation differences at satellite tracks cross-overs acquired less than 15 days apart, in order to limit the impact of potential temporal changes.
5) The spatial uncertainty component for IS2 is calculated in a similar manner to that for CS2: We investigate the relationship between the terrain roughness and spread of elevation anomaly distributions. Based on this relationship, we construct a linear model between roughness and spatial uncertainty.
6) The temporal component is estimated based on a data set of average summer trends of surface elevation (Slater et al, 2021), multiplied with the difference between time of data acquisition and mid-summer. We take this to be an approximate value for $\sigma_{temp}$.
7) We use cross-overs from all years (previously only data from 2019 was used)

Regarding 4), it is worth mentioning that the relationship between terrain roughness and spatial variability is found to differ significantly between the CS2 and IS2 data sets, and we can therefore refute the proposition that a universal relationship should exist between the two.

For IS2, we find a linear relationship between spatial variability and terrain roughness, whereas for CS2 the uncertainty is linearly correlated with the logarithm to terrain roughness (see the updated Figure 2 below, which now includes the IS2 uncertainty model). So, while we now apply a similar scheme to derive the CS2 and IS2 spatial uncertainties, the resulting two uncertainty models differ significantly, with the associated uncertainties consistently being largest for CS2.

We interpret the leveling off of the CS2 uncertainties at high roughness values as being due to the preferential sampling of CS2 at high points in the landscape (POCA), which will lead to a general under-estimation of the observation uncertainty due to spatial variability, particularly in highly irregular terrain. The IS2 instrument, on the other hand, does not display preferential sampling. We observe that the CS2 measurement variability is consistently larger for CS2 than for IS2 at all roughness values.

In general, for both instruments, the temporal uncertainty is notably less important than the spatial uncertainty component, particularly evident in the case of CS2. However, for individual observations the temporal uncertainty component may be non-negligible. We note that with the new uncertainty estimates, the total uncertainties are generally slightly lower than before.

Sections 4.1.1-4.1.2 have been thoroughly revised accordingly. Figure 2 is expanded to also include the IS2 spatial uncertainty model, and a new Figure 3 is included showing the distribution of the individual uncertainty components for the input data. See the updated figures below.

[Figure]

*Figure 2: a, c) Distribution of measured elevation differences (dh) at all CS2 (a) and IS2 (c) satellite track near-cross-over locations within the study area (grey), and cross overs in areas with roughness higher (red) or lower (blue) than the median roughness value of 2.8, respectively. b, d) For CS2 (b), a linear relationship (dashed grey line) exists between the logarithm to the local 100m-scale roughness value and the estimated spatial uncertainty based on the dispersion of associated elevation differences (Eq. 1). For IS2 (d), on the other hand, we observe a linear correlation directly between the roughness and estimated spatial uncertainty. Data from summers 2019 through 2022.*

[Figure]

*Figure 3: Distribution of derived total uncertainties (grey), along with the individual spatial (blue) and temporal (green) uncertainty components for the CS2 (a) and IS2 (b) altimetry data, respectively. Data from summers 2019 through 2022.*

8) Better constrain the variability of CS2/IS2 error (which might explain Fig. 8 entirely).

We have now improved the uncertainty model of the CS2 and IS2 data points, as described above. However, despite the enhancements in uncertainty estimation, limitations persist within the reported error estimates, as visible from Figure 8 (now Figure 9; see the updated figure in the previous comment).

We suspect that the main reason for this discrepancy lies within our estimation of the spatial uncertainty component: When constructing the spatial uncertainty component from temporally close cross-overs, the resulting elevation difference distributions (which we assume describe the observation uncertainty) are not gaussian distributed, but have much heavier tails. In other words, outliers are much more likely to exist that we expect from a gaussian distribution. Nevertheless, as kriging requires gaussian data uncertainties, we approximate these distributions as gaussian. Consequently, we are not properly assigning these extreme data observation uncertainties.

I was happy seeing Equation 1, and then disappointed that the sources were not actually individually refined. I do understand that separating the sources is not feasible with the available data, but it is still possible to separate the variability in error mixed within your single error proxy (here, your cross-over differences).

The authors have 40,000 CS2 cross-over and 60,000 IS2 ones at disposal, a good sample to understand different variabilities. For the temporal variability: the authors could bin these differences with the temporal lag to your reconciled summer date represented by your predicted elevation (middle of your June 1st to September 30th period? This "middle reported date" absolutely needs to be reported in the paper). This should allow to constrain an average variability with the time lag (e.g., Hugonnet et al. (2021), Extended Data Fig. 5ab, which also includies spatial correlations). However, I suspect this variability will likely depends on thinning at the surface at each location, not only on the time lag. You would have to use an estimate of long-term dhdt at the surface (doesn't need to be super resolved, e.g., Smith et al., 2020, or your own product for multiple years) to combine this into a 2-D (time lag + dhdt) binning. This might explain most of your temporal (and general) error variability.

As suggested by the reviewer, we now separate the uncertainty component into its spatial and temporal components.

We investigated several approaches for estimating the uncertainty caused by the temporal variability, but it is hard to disentangle from the spatial variability. The issue is complicated by the following factors:

- The temporal variability tends to be largest at the margin, where also the spatial variability is largest. The two are therefore not independent, and the spatial component is generally the most important factor. Only considering areas with low spatial variability, i.e. primarily central areas, will not accurately capture the temporal variability outside this area.
- The surface elevation displays a trend during summer, the general trend of which has been mapped by Slater et al. (2021). This seasonal trend is very different from the long-term dh/dt trends as in e.g. Smith et al, 2020.
- At any location the surface is not likely to display a constant linear trend throughout the summer, with periods of melting being interrupted by periods of increasing surface elevations due to e.g., new snowfall events.

We attempted the suggested approach of binning with regard to time lag in a similar manner as in Hugonnet et al. (2021), but even when performing a 2D binning by also dividing into regions of similar dh/dt (we tested both using summer trends or long-term trends) the results were inconclusive: The variability was generally constant with time lag, instead of increasing with time lag as expected. We suspect that this is caused by our binning of time lags from different time periods during the summer, and that the surface does not display a steady evolution during this period.

We therefore refrain from estimating an average variability with time lag directly from the data. Instead, we take advantage of the Slater, 2021 map of average dh/dt summer trends, and use these to estimate a temporal uncertainty for each observation as follows:

$$\sigma_{temp}(x, y, t) = \frac{dh}{dt}_{Slater}(x, y) * (t - t_0)$$

With $t_0$ being the reconciled mid-summer date. This date (August 1st) is now reported in the paper.

We further tested the resulting uncertainties at cross-overs to check their consistency: Z-scores were computed from the observed elevation differences divided by the total uncertainty of the elevation differences. Despite heavy tails in the resulting distribution of z-scores, indicating the existence of outliers not fully captured by the estimated uncertainty, the distributions were found to adequately approximate a standard normal distribution. The distributions best approximate a normal distribution for cross-overs located in areas where the Slater et al (2021) map of summer dh/dt were defined, and worse for areas where the temporal component was estimated based on the mean summer trend for the entire ablation zone. The difference was particularly important for the IS2 observations, which to higher degree are located along the ice sheet margin, and therefore outside the Slater data region.

**Main comment 2: Report spatial correlation of errors for your data product**

As pointed out by the authors in the short discussion (L617-620), even though your estimates have a reasonable error, it might not be random at all because of the temporal differences in acquisition.

While it would be quite a task to correct these temporal errors in detail (feasible by extending your kriging framework with a dimension of time or seasonality, for instance, or with a more simple models of your multi-annual changes during the summer season), the authors can still model the spatial structure of these errors. And this information would be quite important for downstream applications.

As noted by the reviewer, the uncertainties of the PRODEM product are not random, but display systematic deviations due to the spatial correlations of acquisition time of measurements, with one satellite track being measured at the same time. We agree that it could be interesting to extend the kriging framework with a time dimension to fully account for these seasonal changes. However, as also pointed out by the reviewer, this would be quite a task, which we consider to be out of scope for the current paper.

Instead, we follow the reviewer's suggestion to model the spatial structure of errors for the PRODEMs, as described below.

At the local scale, those are systematic errors due to common temporal sampling. But, as long as there is no bias in the time of data acquisition, those will be centered on 0 on average (which the authors validate with their comparison to IS2 in a LOOCV, Table 1). Which means that, at the larger scale of Greenland, they can be interpreted as structured errors with spatial correlations due to the temporal sampling discrepancies (Hugonnet et al. (2021)).

Here, the authors should at least report an "average" variogram that represents the spatial correlation in errors their products due to temporal sampling (this has be done on the LOOCV differences you compute in Section 7.1).

As part of the validation of the PRODEMs, we have now added a section describing the spatial structure of correlated errors based on the normalized prediction errors from the LOOCV. Since the error structure is likely to be aligned with the satellite tracks, in particular the IS2 observations forming straight lines, we construct directional variograms along and across track directions. In the along-track direction, the error

structure has an effective correlation length of a few kilometers, whereas errors in the across-track direction are essentially uncorrelated.

We further provide an analysis of the prediction errors (raw and normalized) stratified relative to surface roughness, elevation, latitude, ArcticDEM uncertainty etc., allowing us to identify troublesome areas for the predicted elevation and reported uncertainties.

Even better would be to have a varying variogram depending on a map of "time lag to closest elevation used" or similar, that would reliably represent the errors in both space and time!

You could then validate your variogram representing your errors using Block LOOCV, and studying that the spread matches that predicted by your variogram in space (following classical error propagation with a spatial correlation term, e.g. Hugonnet et al. (2022), Equation 18).

However, the "swath" nature of your input data, and hence the temporal sampling, is less than ideal to capture errors in temporal sampling with omnidirectional variograms. It's still an important added value, and I don't see an easy solution to this directional problem, except trying to come up with a better scheme to correct these temporal errors seasonally to your middle summer date, using your own multi-annual data (as stated at first).

As previously mentioned, we now provide a better estimate of the input data uncertainty of representing the mid-summer elevation surface due to their different acquisition time.

While we agree with the reviewer that it would be ideal to construct the PRODEMS using a 3D kriging framework to fully account for the seasonal changes, we consider this to be out of scope for the current paper – but we appreciate the comments, and would like to work on improving this aspect for future editions of the PRODEMs.

**Main comment 3: (Optional) Regionally-varying kriging: could also decompose and standardize your elevation field variance sources to explain the variability at the source**

I must have been quite the effort to setup this regionally-variable scheme with GSTools!

First, while the Cressie estimator is reasonably robust to outliers, it is still fairly sensitive for what we get in elevation data (ArcticDEM outliers). You could consider using Dowd's estimator (Dowd, 1984) that actually matches your NMAD in terms of how it is derived. This estimator is implemented in SciKit-GStat (Mälicke et al., 2022) which has functions to export models to GSTools.

We now produce the experimental variograms using Dowd's estimator. Consequently, we are now using SciKit GStat for experimental variogram estimations, instead of GSTools where this estimator is not implemented. We only use the tool to produce the experimental variograms.

Second, while the regionally-varying kriging is well-implement and definitely valid, it might be over-fitting your data locally by using that regionalized variogram subsample. This would result in an over/under-smoothing of the final interpolated field compared to reality, because of the way the scheme is implemented. Another way to avoid the non-stationarity would be to describe the nature of your spatial covariance and its variability (summed variance components with a certain correlation range, and their individual variability). The variability in error with long correlation range would for instance likely linked to time lag and dhdt (Main comments 1 and 2). Then, using standardization + decomposition (as in done a bit in Hugonnet et al. (2022) with instrument resolution and noise, but was less important there), you could transform your data to reach second-order stationarity, and apply kriging.

I am mostly presenting this option for the authors to realize that there are ways to avoid a regionally-varying variogram implementation. But I think that, given the complexity of this scenario (varying time

sampling that is uncorrected), the current approach is completely valid and it would be hard to implement a decomposed + standardized approach.

We appreciate the suggestion. We did at first try to implement a standardization scheme, but were not successful in making it work properly, and hence decided to go to regionally varying kriging approach. One likely issue for implementing the standardization scheme was that ArcticDEM used as reference DEM had different properties in different regions; with the new version of ArcticDEM this may (or may not) be less important.

**Main comment 4: Justify the choice of 500 m for the final product**

I don't think it is explained anywhere in the manuscript why the authors chose 500 m for their study, it would be good to add a paragraph on this!

We have added the following paragraph on why we use 500m resolution for the final product:

*The PRODEMs are constructed using Point Kriging, in which approach the interpolated elevation anomaly field is evaluated at the grid cell centres. For this approach to accurately represent the mean field value within a grid cell, the grid resolution must align with the scale of variability of the observation data. After down-sampling of the IS2 data, both CS2 and IS2 data sets are representative for an area of a few hundred meters, and an appropriate resolution for the anomaly field of the interpolated PRODEMs is therefore on the order of 500 m.*

Additionally: ArcticDEM mosaic v4 is released since last month, with significant improvements. If it is not too much work, the authors could update their pipeline with this latest version.

This has been updated, and the text and figures throughout the paper have been changed accordingly.

We note that we observe a checkerboard pattern in dh also when using the new version of the ArcticDEM as reference – although less pronounced than previously, and appearing more significant in other areas. The updated Figure 10, from a different location, is provided below.

[Figure]

**Figure 10: Elevation anomalies (here for PRODEM19) relative to ArcticDEM display a checkerboard pattern due to the way ArcticDEM was constructed. Two ArcticDEM tiles of 100x100 km are indicated.**

**Main comment 5: Drop the nugget for the kriging**

Using a nugget in a variogram represents a very specific type of discontinuous spatial pattern at the boundary of the measured location. This is really only relevant to point observations that actually belong to a small spatial ensemble (such as gold nuggets), and is not relevant to inherently continuous elevation data on a grid.

You can actually see on Fig 4. the misfit of the black and red variograms at short spatial lags: they should not have a nugget at all, and it would show if more short lags were sampled.

We do not agree with the reviewer that the nugget should be dropped completely - in our case, the nugget should reflect the observation uncertainties. However, as the reviewer points out, during modelling of the variogram, the nugget value is often overestimated. We therefore now prescribe the nugget value as the median of the uncertainties of the surrounding observations when constructing the variograms.

However, this change gave rise to several other changes in the variogram estimation procedure. With a prescribed nugget value, it was often not possible to construct well-fitting variograms out to a distance lag of 15km. Since we only use the closest 200 points for interpolation, we only need to know the variogram structure at relatively close distances. We therefore changed the variogram binning, decreasing both the bin size and the maximum bin size. With this adjustment, the misfit of the variograms often improved at small spatial lags.

Constraining the nugget value also meant that the R2-score no longer was an appropriate measure to describe how well the variogram model fitted the experimental variogram: In some locations, the calculated semi-variance at the first lag (and first lag only) was substantially larger than the prescribed nugget value (as well as the semi-variance at the subsequent lags). For some of these, the R2-value became negative despite a visually good fit. We therefore now refrain from using the R2-score to filter out outliers in the variogram parameter maps.

To improve your spatial lag sampling in the variograms, consider using a non-uniform binning. Those can be long to compute and, for pairwise sampling efficiency, in SciKit-GStat, there are different types of metric spaces just for this, one based on the sampling scheme presented in Hugonnet et al. (2022), Fig. S13. Those are also wrapped more conveniently in xDEM for raster data (xDEM contributors, 2022).

We agree that this is something to consider for future PRODEM versions. For now, however, we have decided to go a slightly different way. When modelling the experimental variogram, we now weight the value at a given lag with the number of pairs of observations within the lag. In this way, a well-defined semi-variance value is weighted higher when fitting the model. While incorporating this, we adjusted the weighting to decrease over 1km instead of 5km (L374), as this (due to the overall spatial distribution of observations) resulted in an approximately similar weighting scheme as previously.

**Line-by-line comments:**

Title: Having "present" in the title for the period is probably not a good idea long-term.

We have changed the title to "(2019 through 2022)" for improved clarity.

L40-42: This statement unlikely holds true. Typical random errors in modeled ice thickness are +/- 100 meters near flux gates with important spatial correlations as well as systematic errors (Kochtitzky et al., 2022; 2023). While predicted surface elevation errors are typically less than 10 meters (Hugonnet et al., 2021). Even measured (not just modelled) ice thicknesses are usually less precise and with larger correlations in errors than measured surface elevation. The authors could simply make the case that we need more annual data to represent the surface elevation of the ice sheet.

This has been removed, and we have changed the introduction so that it now focuses more on the importance of ice sheet elevation monitoring, and how the PRODEMs will offer new insight into ice sheet monitoring efforts.

L48-52: Same remark.

See answer above.

L81 and onwards: Need to add spaces between numerical values and unit.

Done.

L190: Could directly introduce the Normalized MAD or "NMAD" directly, which has been around for while in elevation data analysis (e.g., Höhle and Höhle, 2009).

Agreed. We now write as follows:

*While not strictly adhering to a Gaussian distribution, an approximate Gaussian standard deviation describing the spread of the distribution can be obtained using normalized MAD: $s_{MAD} = 1.4826 * MAD$. This approach has the advantage of being more robust to outliers than a direct calculation of the standard deviation.*

L319: Need to specify: "second-order" stationary!

Done

**References:**

Mirko Mälicke, Romain Hugonnet, Helge David Schneider, Sebastian Müller, Egil Möller, & Johan Van de Wauw. (2022). mmaelicke/scikit-gstat: Version 1.0 (v1.0.0). Zenodo. https://doi.org/10.5281/zenodo.5970098

Hugonnet, R., Brun, F., Berthier, E., Dehecq, A., Mannerfelt, E. S., Eckert, N., & Farinotti, D. (2022). Uncertainty Analysis of Digital Elevation Models by Spatial Inference From Stable Terrain. *IEEE Journal of Selected Topics in Applied Earth Observations and Remote Sensing*, *15*, 6456–6472.

Hugonnet, R., McNabb, R., Berthier, E., Menounos, B., Nuth, C., Girod, L., Farinotti, D., Huss, M., Dussaillant, I., Brun, F., & Kääb, A. (2021). Accelerated global glacier mass loss in the early twenty-first century. *Nature*, *592*(7856), 726–731.

Kochtitzky, W., Copland, L., King, M., Hugonnet, R., Jiskoot, H., Morlighem, M., Millan, R., Khan, S. A., & Noël, B. (2023). Closing Greenland's mass balance: Frontal ablation of every Greenlandic glacier from 2000 to 2020. *Geophysical Research Letters*, *50*(17). https://doi.org/10.1029/2023gl104095

Kochtitzky, W., Copland, L., Van Wychen, W., Hugonnet, R., Hock, R., Dowdeswell, J. A., Benham, T., Strozzi, T., Glazovsky, A., Lavrentiev, I., Rounce, D. R., Millan, R., Cook, A., Dalton, A., Jiskoot, H., Cooley, J., Jania, J., & Navarro, F. (2022). The unquantified mass loss of Northern Hemisphere marine-terminating glaciers from 2000–2020. *Nature Communications*, *13*(1), 1–10.

Höhle, J., & Höhle, M. (2009). Accuracy assessment of digital elevation models by means of robust statistical methods. ISPRS Journal of Photogrammetry and Remote Sensing: Official Publication of the International Society for Photogrammetry and Remote Sensing , 64(4), 398–406.

xdem contributors. (2021). xdem (v0.0.2). Zenodo. https://doi.org/10.5281/zenodo.4809698

---

## Author Comment (AC2)

**Replies to reviewer #2**

We thank the reviewer for their constructive comments and have addressed the points raised as described in the following, with particular focus on clarifying the kriging procedure and underlying assumptions. The reviewer comments are shown below in blue, and our replies shown in black. Line numbers refer to the original version of the manuscript that has been reviewed.

Since the manuscript was submitted, several of the input datasets have been updated:

- We use CryoSat-2 Baseline E for all years, as it is now available for the entire period
- ICESat-2 ATL06 has been updated to version 6; the new version has corrected the geolocation error found to exist in version 5
- ArcticDEM mosaic has been updated to v4.1 (as also proposed by the reviewer)

We have produced a new version of the PRODEMs using the updated data sets, and the text and figures in the manuscript have been adjusted accordingly. This includes e.g., an updated version of figure 10 now displaying another area that shows a similar checker-board pattern in elevation anomalies relative to ArcticDEM v4.1 (albeit less pronounced than previously found when comparing to ArcticDEM v3), and removal of figure 11 (including the related discussion) which is no-longer relevant.

We hope that with these and the below changes, you will accept to review an updated version of our paper.

This paper introduces a set of four 500-meter resolution annual Digital Elevation Models (DEMs) representing the Greenland ice sheet marginal zone during the summers of 2019 to 2022, referred to as PRODEMs. Covering a 50km wide band from the ice edge, these DEMs meticulously encompass all outlet glaciers of the Greenland ice sheet. The integration of DEMs derived from ICESat-2 and CryoSat-2 constitutes a commendable initiative, providing an additional valuable source of topographic information for Greenland. While the paper is well-crafted overall, there are certain statements that require precision to ensure accuracy and clarity. It is crucial to address these points before considering the paper for publication.

**General comments:**

- The authors' decision to generate the DEM specifically for the summer and focus on the marginal zone of the ice sheet raises pertinent questions. What rationale underlies the choice of season, and what significance does the selected ice sheet margin hold in this context? Furthermore, a clarification regarding the specific function of the DEM under these conditions would enhance the paper's comprehensibility.

This is described in L114-120. In short, we wish to develop the DEM for areas (and periods) where the surface is minimally covered by snow since the two sensors (CS2, IS2) interact differently with the snow. This leaves the marginal zone during summer, with summer being the period with least snow cover, which we here take to be June through September. The selected area encompasses most of the summer bare ice zone, but to avoid a product with sparse and/or intermittent coverage, and to maintain the same coverage across years, it was decided to cover the entire 50km margin of the ice sheet.

To clarify this to the reader, we now also mention this earlier in the manuscript (L44):

*Area and time period are selected to obtain areas minimally covered by snow, for which the two satellite sensors are expected to measure the same surface.*

The resulting DEM is therefore the ice surface, without snow cover, which is e.g. the DEM appropriate for mass balance considerations in the ice sheet marginal areas. This has been clarified in the paper (L46):

*The PRODEMs therefore represent the ice surface topography, which is e.g., the DEM appropriate for marginal mass balance considerations.*

- The authors employ two types of altimeter data for DEM generation, a method requiring careful consideration of their consistency. While the authors assert that the disparity between the two datasets is negligible, it is crucial to acknowledge that applying these datasets without consistency correction may influence elevation estimations. Particularly noteworthy is the comparison between IS2 and CS2; the latter exhibits uneven spatial distribution and coarse resolution, introducing potential biases in the generated DEM. My concern centers on the relatively lower measurement accuracy and disparate spatial distribution of CS2, aspects that warrant careful attention.

We agree with the reviewer that it is most important to account for potential biases when combining information from the two types of altimeter data. In this respect, we emphasize that because of our choice of region and period (summer-time margin only), it is believed that there should be no elevation bias between the two altimeters, as supported by existing studies and well documented in the literature. We now refer to these studies in the text (L275):

*The PRODEMs, however, are constructed based on elevation data from the ice-sheet marginal areas during summer, where the snow cover is limited. In the bare ice zone, it is well documented in the literature that the elevation bias of radar altimeters is negligible (e.g., (Dall et al., 2001; Otosaka et al., 2020; Davis and Moore, 1993)), and we therefore expect small biases only between the employed CS2 and IS2 elevations within the PRODEM area.*

Nevertheless, we cautiously conducted a thorough analysis of potential biases by comparing elevations measured at crossing satellite tracks of the two satellites, and we found that (as expected) the disparity is negligible. Indeed, not even the sign of the median of the elevation differences is consistent among basins (L280-281).

As the reviewer correctly points out, the CS2 data is associated with lower measurement accuracy than IS2. During construction of the PRODEMs, we account for this by assigning uncertainties to the individual observations based on the sensor type (as well as location and acquisition time). We have in the updated version of the PRODEMs and manuscript thoroughly revised the uncertainty assessments to now fully consider the uncertainty contribution from spatial and temporal uncertainty for each observation. The updated distributions of uncertainties for the CS2 and IS2 observations are shown in the updated Figure 3, see below. As for the previous PRODEM uncertainty calculations, the uncertainties remain significantly larger for CS2 (median: 0.98 m) than for IS2 (median: 0.21 m).

[Figure]

**Figure 3: Distribution of derived total uncertainties (grey), along with the individual spatial (blue) and temporal (green) uncertainty components for the CS2 (a) and IS2 (b) altimetry data, respectively. Data from summers 2019 through 2022.**

Another aspect brought up by the reviewer is the different resolution of the two datasets, with the IS2 data having much higher resolution than the CS2 data. In order to work with consistent datasets, we down-sampled the IS2 data along track to 250m resolution, which is of similar order as the resolution of the CS2 data (L90). We have added the following (in red) to this sentence:

*For consistency with the resolution of CS2 observations, and given that the topographic variations to be resolved in the 500m resolution PRODEM are much smoother than 20m, we further down-sample the ATL06 data by computing median values over 250m along-track segments of each beam.*

The last aspect of the differences between the two measurement types mentioned by the reviewer is the difference in their spatial coverage: While valid CS2 observations do not exist for the rough terrain often located along the outer margin of the ice sheet, the IS2 radar is able to obtain valid observations also for these areas. Additionally, the IS2 data are acquired along straight lines, whereas the CS2 observations are spatially more scattered. However, these differences in spatial coverage will not affect the interpolation of the resulting DEM, apart from resulting in generally higher uncertainty of the resulting product in areas with less data.

- Regarding data coverage, the authors applied distinct interpolation radii in different regions. To enhance reader understanding, it would be beneficial to include information on data coverage, such as footprint numbers for CS2 and IS2 in each grid. This insight can shed light on the grids effectively covered by the altimeters, offering rationale for the choice of a 500 m resolution. It's noteworthy that the ICESat-2 DEM by Fan et al. (2022), constrained to 10 footprints at 500m resolution, covers only approximately 30% of the entire ice sheet, and a discussion on this limitation could be insightful for the readers.

It is actually not very relevant how many observations are located within each grid cell. Consider, for instance, an area of very flat terrain: It is still possible to interpolate a reasonable elevation value here, even for a grid cell in which no observations exist. The important factor is the number of observations within the correlation length of the varying terrain (i.e. effective length scale of the variogram). If no observations exist within this interpolation radius, the interpolated value will be set equal to the local median field.

We have added a figure to the paper showing the number of observations within the spatially varying effective length scale from each grid cell (see below), along with a discussion. We find that 13% of the grid cells suffer from sparsity in neighboring data, which we take to be less than 5 observations within a radius corresponding to the local effective length scale. And for 7% of the grid cells, no observations exist within the interpolation radius. The issue is most pronounced in the southern regions; for the southernmost

drainage basin (basin 5.0) the percentage with sparse data coverage has increased to 37% (21% without any nearby observations). In general, however, the interpolated field is based on 22 nearby observations (median of the distribution). In regions without any nearby observations, the ArcticDEM elevations are adjusted with a constant depending on the local median anomaly field, as calculated from the 200 closest altimetry observations, and the resulting elevation field is thus highly reliant on ArcticDEM.

[Figure]

An additional parameter that one may use to evaluate the coverage of data points is the distance from grid centers to the closest data point. Histograms with these distributions are found in Figure 1a. When discussing this figure, we now have added a mention of the total percentage of grid cells having the closest data point more than 1 and 2 km away (L129):

*For 19% of the grid cells, the closest observation is more than 1 km away, but for only 4% it is more than 2 km away.*

None of these measures can, however, directly be translated into an appropriate grid resolution, as this depends on the spatial representability of the data. Our rationale for selecting a grid resolution of 500 m is described later in this document.

- The authors should elucidate the rationale behind selecting the Kriging interpolation method. Given the steeper terrain at the ice sheet's edge, it is essential to address whether Kriging interpolation can effectively fulfill the task of estimating elevation in such rugged terrains. Clarification on the suitability of the chosen interpolation method, especially in challenging topographical conditions, would contribute to a more comprehensive understanding for the readers.

Our decision to employ Kriging interpolation was based on several factors, including its ability to account for variable uncertainty of observations, its robustness in capturing spatial variability and its ability to provide smooth and continuous surfaces. We have added the following to L316 (in red):

*Kriging can account for variable uncertainty of individual data points during the interpolation, and due to its robustness in capturing spatial variability it is a widely used method to interpolate geophysical fields (e.g. (Bamber et al., 2001; MacGregor et al., 2015; Bales et al., 2001)).*

In this respect is it very important to note that we do not apply kriging to the original elevation measurements, but to elevation anomalies relative to ArcticDEM. These elevation anomalies display a much smoother field, also around the marginal areas (see figure below), and to large extent this resolves the issue raised by the reviewer of whether kriging can also be used in these areas. Even for areas with very rough surface, such as those often existing at the very outer part of the ice sheet edges, there is consistency in the observed elevation anomalies across length scales of several kms. We have added this argumentation to the paper, along with the figure.

[Figure]

As long as the spatial correlation of the anomaly field can be correctly captured by the variogram, and the field (largely) fulfills the statistical requirement for kriging to be valid, there is no reason that kriging, which offers an optimal unbiased linear interpolation method, should not be a good interpolation method.

- There are reservations regarding the chosen 500m resolution. Satellite altimeters typically yield fewer effective observations at the ice sheet's edge (especially for CS2), and opting for higher resolution may result in a reduced number of usable observation grids. This reduction can potentially impact the interpolation performance. The question arises: Can a DEM heavily reliant on interpolation accurately portray the actual elevations, particularly in the steeper terrains at the ice sheet margins? Additionally, in regions with lower latitudes, where the gaps between satellite orbits are more substantial, concerns linger about ensuring effective spatial interpolation. Addressing these considerations would fortify the paper's discussion on resolution choice and its implications.

As the reviewer correctly points out, there are fewer observations around the very edges of the ice sheet with rough terrain, not least since these are not covered by CS2 data. Again, we are partially saved by using ArcticDEM as a reference DEM, which has high spatial resolution also in these regions. With the anomaly field here often displaying small correlation lengths, the resulting PRODEM tends to heavily rely on the ArcticDEM topography, the latter being slightly adjusted by the median value of the local anomaly field. But, agreed, the PRODEMs do have much larger uncertainty in these areas of very difficult terrain.

As previously mentioned, the spatial density of observations cannot directly be translated into an appropriate grid resolution, which is heavily dependent on the spatial representability of the data.

In terms of the chosen 500 m resolution: We have chosen to use Point Kriging, instead of Block Kriging, to obtain the elevation field, which means that the interpolated field is evaluated at the center of each grid cell. To improve the estimation of the mean value of the grid cell, we subsequently smooth the derived elevation field using a 3x3 mean filter. When taking the Point Kriging approach, it is important that the input data used for the interpolation is representative for the elevation field at the scale of the grid cell. Consider, for instance, the example of using highly resolved data (e.g. using only IS2 data without down sampling) with one observation located close to the center of the grid cell: The center value will then largely reflect the elevation of the nearest data point, regardless of whether this value is representative for the entire grid cell. In smooth areas, this may not be an issue, but in areas of rough terrain, this may not be very accurate. However, if the observation close to the grid cell center is representative of an area of approximately the same size as the grid cell, the interpolated value will correctly represent the mean grid cell value.

As the resolution of the CS2 is on the order of ~500 meters, we choose to down sample the resolution of the IS2 data to similar resolution (we decided to down sample to only 250 m along track to increase the data density), and, consequently, the appropriate resolution of the associated anomaly field must be chosen to be on the same order, namely 500 m.

We have added the following paragraph to the paper to clarify the chosen resolution of the product:

*The PRODEMs are constructed using Point Kriging, in which approach the interpolated elevation anomaly field is evaluated at the grid cell centres. For this approach to accurately represent the mean field value within a grid cell, the grid resolution must align with the scale of variability of the observation data. After down-sampling of the IS2 data, both CS2 and IS2 data sets are representative for an area of a few hundred meters, and an appropriate resolution for the anomaly field of the interpolated PRODEMs is therefore on the order of 500 m.*

- Notably, the entire DEM is subjected to Kriging interpolation, implying a potentially limited actual coverage area by the altimeter. Therefore, it is imperative for the authors to furnish the proportion of space covered by observational data. Elevation derived from Kriging interpolation might deviate from the altimeter-observed elevation, necessitating consideration of this difference, as the potential errors induced by interpolation are a critical aspect. Particularly, given the uneven spatial distribution of CS2, which may introduce additional uncertainty to Kriging interpolation, it becomes crucial to address how the authors accounted for the impact of observation gaps on the results. Discussing these considerations would enhance the paper's transparency regarding the intricacies of the interpolation process and associated uncertainties.

As previously mentioned, the important factor is the number of observations within the correlation length of the varying elevation anomalies, and we now provide a figure showing this distribution. In areas poorly covered by the observations, the resulting surface is highly reliant on the ArcticDEM surface topography used as reference, and the uncertainty of the interpolated surface is increased.

With proper observation uncertainty estimation and spatial variability model (aspects that we thoroughly investigated) kriging is a way to reconcile the surrounding observations according to their uncertainties and spatial covariance. For it to provide good measurements of the mean field value at grid cells, it is, as mentioned previously, furthermore imperative that the grid cell size is aligned with the spatial representability of the observations. As long as these (and other) conditions for kriging to perform well are fulfilled, the interpolated surface should only deviate from observations within their associated uncertainties.

In terms of the mentioned un-even spatial distribution of CS2, we remark that actually the more "random" distribution of CS2 is more appropriate for interpolation by kriging than the straight lines of IS2 observations (which are far from randomly distributed).

We have added the following paragraph (L333) to clarify the prerequisites and assumptions behind using kriging interpolation:

*The accuracy of the interpolated field will be limited if the number of sampled observations (within the effective interpolation radius) is small or not adequately distributed relative to the properties of the field. The accuracy of the PRODEM elevation anomaly field will therefore be reduced in areas of high variability and few observations, such as conditions prevalent at the outer edges of the ice sheet.*

- Addressing DEM uncertainty requires a more detailed exposition. While the spatial and temporal uncertainty is elaborated upon, clarification is needed on how the authors define the instrument and geolocation uncertainties mentioned in equation (1). It would be beneficial to delineate the

*specific contributions of each factor to the overall uncertainty and provide a proportionate breakdown.*

We have thoroughly revised the procedure of addressing the CS2 and IS2 uncertainty in the following ways:

1) We now separate the uncertainty component into its spatial and temporal components
2) Elevation differences are now computed as the difference of elevation anomalies (instead of elevations); this approach accounts for a potential slope in the area.
3) We reduce the allowed spatial distance between measurements to 50m (instead of 100m)
4) The spatial component is computed only based on elevation differences at cross-overs acquired less than 15 days apart, in order to limit the impact of potential temporal changes.
5) The spatial uncertainty component for IS2 is calculated in a similar manner to that for CS2: We investigate the relationship between the terrain roughness and spread of elevation anomaly distributions. Based on this relationship, we construct a linear model between roughness and spatial uncertainty.
6) The temporal component is estimated based on a data set of average summer trends of surface elevation (Slater et al, 2021), multiplied with the difference between time of data acquisition and mid-summer.
7) We use cross-overs from all years (previously only data from 2019 was used)

Definitions of the various uncertainty components are provided in L156-163. Due to the way that we derive the spatial uncertainty component, this component includes the geolocation uncertainty, as well as the measurement uncertainty, and this is now clarified in the manuscript:

*In the following, we develop separate uncertainty models for the two data sets to estimate representative values for their spatial uncertainty (including effects from geolocation and instrument measurement uncertainty), as well as a common model for the temporal uncertainty.*

At small roughness values, we can evaluate the instrument measurement uncertainty from the combined spatial uncertainty as 40cm (CS2) and 8cm (IS2), respectively. These values are now mentioned in the text. We cannot reliably disentangle the geolocation uncertainty from the uncertainty due to spatial representability, as these are intricately intertwined.

We further elaborate on the relative importance of the two terms (spatial vs. temporal):

*In general, for both instruments, the temporal uncertainty is notably less important than the spatial uncertainty component, particularly evident in the case of CS2. However, for individual observations the temporal uncertainty component may be non-negligible, with 21% of the IS2 observations having larger temporal than spatial uncertainty. For CS2, this is only the case for 0.4% of the observations.*

Moreover, in discussing the uncertainty of CS2/IS2 elevations, the term 'crossovers' might be prone to misunderstanding.

We have gone through the manuscript and changed the word "cross-overs". We now refer to it as "observations near intersecting satellite tracks".

Consider utilizing track-based crossover analysis for data validation, ensuring a clearer understanding of whether the differences approach zero. This adjustment would enhance precision in evaluating the uncertainty associated with CS2/IS2 elevation comparisons.

We have followed the reviewer's suggestion, and tested the resulting uncertainties at intersecting satellite tracks to check their consistency: Z-scores were computed from the observed elevation differences

divided by the total uncertainty of the elevation differences. Despite heavy tails in the resulting distribution of z-scores (indicating the existence of outliers not fully captured by the estimated uncertainty) the distributions were found to adequately approximate a standard normal distribution.

For some marginal areas, the temporal component was not well defined due to lack of data, and for these areas, we used the average value of trend of summer elevation across summer to estimate the temporal representability. The z-score distributions best approximate a normal distribution for cross-overs located in areas where the temporal component was best defined. This issue is particularly important for the IS2 observations, which to higher degree are located near to the ice sheet margin, where the temporal component is poorly constrained.

We have added the following paragraph to the paper:

*The derived observation uncertainties were subsequently tested for consistency with the observed elevation differences near intersecting satellite tracks. Z-scores were computed from the observed elevation differences divided by their total uncertainty. Despite heavy tails in the resulting distribution of z-scores, indicating the existence of outliers not fully captured by the estimated uncertainty, the distributions were found to adequately approximate a standard normal distribution.*

- The evaluation of the proposed DEM solely through cross-validation may be deemed insufficient. It is recommended that the authors endeavor to gather actual measurements, such as airborne datasets, for Greenland elevation. This would facilitate an independent evaluation of the generated DEM, offering accuracy metrics under various terrain conditions. Such an approach would provide more robust and precise evaluation information, enhancing the credibility of the study's findings.

We agree that it would be great to also validate the PRODEMs against independent data. However, airborne data sets that fulfill the requirements of being acquired during the summer months of 2019-2022 are exceedingly sparse. To our knowledge, only one such campaign exists, namely the CryoVEx Summer 2019 campaign, and it was directed towards obtaining data over sea ice rather than land ice. One track does, however, cover a small section of the very margin of the north-eastern part of the ice sheet. We will investigate whether this track contains sufficient land ice data for independent validation (which in any case will be very limited), and if so, we will report the results in the manuscript.

As this data set in any case is too limited to offer accurate evaluation metrics under diverse terrain conditions, we instead stratify the prediction errors from the cross-validation exercise, and consider the error distributions as function of increasing latitude, elevation, roughness, aspect, ArcticDEM uncertainty and summer elevation trends. We have added a section on this to the manuscript, along with the figure provided below.

In short, the analysis allows us to identify troublesome areas, where one should be careful regarding the reported uncertainties in the PRODEMs, these being characterized as having $s_{MAD}$ values of the normalized prediction uncertainties significantly above one, which we here take to be larger than 1.2. In order of importance, these areas are characterized by having: Roughness above 13 m, ArcticDEM MAD values above 0.4 m, and/or elevations below 300 m. These conditions span respectively 25, 22, and 4% of the area, and largely cover the same areas along the very margin of the ice sheet.

[Figure]

**Figure 11: Descriptive measures (median and $s_{MAD}$) of the distribution of prediction errors, both raw (dashed lines) and normalized errors (solid lines), when stratified relative to various location and terrain properties that may impact the PRODEM product performance. Bins are created to cover the 5 to 95 percentiles of each data set. The ArcticDEM MAD value is computed as the median absolute deviation of the DEMs contributing to ArcticDEM at a given location, and therefore provides an uncertainty estimate of ArcticDEM. Data based on prediction errors from PRODEM19.**

- In comparing with other DEMs, presenting a distribution map alone may lack depth. Additional comprehensive comparisons, such as showcasing the spatial distribution or numerical values of elevation differences, and establishing relationships between these differences and slope and roughness, are warranted. To enhance the display of the DEM, the authors might consider incorporating a slope map or shaded relief map. These additions would provide a more nuanced evaluation of the DEM data and contribute to a thorough understanding of its effectiveness.

We have added a slope map to Figure 9 (now Figure 10), along with elevation difference maps relative to IS2DEM19 and ArcticDEM, and histograms of the associated distributions, see the updated figure below. Since the PRODEMs have much more structure (as evident, for instance, from the slope map), we also compare it to a smoothed version of IS2DEM19.

As the three DEMs are not representing the same period of time, and the IS2DEM19 is of lower actual resolution than the others, we cannot use the DEM comparisons for validation of PRODEM19. We

therefore do not consider it relevant to dive further into the investigation of relationships between elevation differences and e.g. slope and roughness.

[Figure]

**Figure 10: A comparison of the three elevation fields: IS2DEM19, PRODEM19, and ArcticDEM across Petermann Gletsjer. Coverage of the three DEMs varies slightly across the region. The top panel shows elevation differences between the three DEMs, including a filtered version of PRODEM19, produced using a Gaussian filter with 5 pixels radius and relative standard deviation of 50%. The middle panel shows the elevation fields across Petermann Gletsjer. The lower panel compares the slope distribution for the three elevation models. a) A histogram of the distribution of elevation differences across the PRODEM area, and b-d) maps showing the spatial distribution across Petermann Gletsjer. e-g) The elevation fields across Petermann Gletsjer. h) Distribution of slopes across the PRODEM area, i) mean and median values of the slope distribution, and j-l) the spatial slope distribution across Petermann Gletsjer.**

- Consideration should be given to the potential benefits of updating the DEM with the new ArcticDEM mosaic v4, as it has the potential to significantly enhance DEM accuracy and reduce differences in dh (as observed in Figure 10). Such an update could serve as a viable solution to the issues identified and contribute to an overall improvement in the quality of the DEM.

The PRODEMs have been updated used the new ArcticDEM v4.1 mosaic. We note that we observe a checkerboard pattern in dh also when using the new version of the ArcticDEM as reference (although less pronounced than previously, and appearing more significant in other areas). The updated Figure 10 (now Figure 11), from a different location, is provided below.

[Figure]

**Figure 11: Elevation anomalies (here for PRODEM19) relative to ArcticDEM display a checkerboard pattern due to the way ArcticDEM was constructed. Two ArcticDEM tiles of 100x100 km are indicated.**

**Specific comments:**

(p: page, l: line)

Section Introduction: The current paragraph lacks substantial content. To enhance its depth, it is recommended to incorporate the significance of ice sheet elevation monitoring. Elaborating on the importance and implications of monitoring ice sheet elevation would provide readers with a clearer understanding of the broader context and relevance of the study.

We have revised this section so that it to higher degree focuses on the importance of ice sheet elevation monitoring.

p2l44, Supplementing the existing DEM datasets and providing commentary on them is advised. This additional information will not only enhance the comprehensiveness of the study but also offer valuable insights and context for readers evaluating the significance and limitations of the proposed DEM datasets.

In the introduction to the revised manuscript, we now also describe existing DEM datasets, and describe how the PRODEMs will offer new insights into ice sheet monitoring efforts.

p2l53, CS2 is equipped with a Ku-band radar altimeter, which is highly sensitive to moisture changes. Notably, surface melting can introduce significant interference to the echo signal, posing challenges in obtaining accurate elevations during such periods. While the author's emphasis is on capturing surface elevations during summer, it is worth considering the possibility of melting snow, even in areas designated as snow-free. Clarification on the impact of any residual melting snow on elevation data would enhance the understanding of potential influencing factors.

The effect of surface melting on the radar signal properties is primarily relevant in the firn zone (i.e. outside the PRODEM area), where surface melting will affect the structure of the near-surface firn, this causing significant biases in the elevation estimates.

Indeed, the ice sheet summer surface in the marginal areas is inhomogeneous (experiencing surface melting surface, the existence of supraglacial lakes, may at times be partly snow-covered etc.), which may impact the altimetric measurements of the ice sheet surface, as it may make the surface less well-defined and thereby more difficult to track in the radar waveform. However, we consider the potential impact of this on the elevation estimates to be well within the uncertainties of the CS2 observations.

To eliminate bad CS2 measurements (due to this factor as well as e.g., incorrect geolocation), we filter the altimetry observations prior to interpolation.

P2l69, what distinguishes the Baseline-D and E datasets, and is there potential for this difference to impact the consistency of the CS2 data? Clarifying the distinctions between these datasets and addressing their potential influence on CS2 data consistency would contribute to a more comprehensive understanding of the data sources employed in the study.

Since submission of the paper, the baseline-E data set has become available for the entire period, and we have therefore produced a new version of the PRODEMs using the baseline-E for the entire period, so the CS2 data set is now fully consistent.

P2l74, in contrast to IS2, CS2 exhibits a relatively limited dataset with irregular coverage, further reduced through quality control measures. It is imperative to elucidate the specific spatial distribution of CS2 data after quality control. Any potential uneven distribution in the data may raise concerns about its impact on subsequent analyses and differences. Addressing these considerations would provide a clearer understanding of the reliability and potential biases associated with the quality-controlled CS2 dataset.

The CS2 datasets constitute ~25% of the full dataset applied to construct the PRODEMs (L93).

For best performance of the kriging interpolation, the observations should be distributed in an adequately balanced way with respect to the properties of the field to be interpolated. In this respect, the CS2 data is actually much better distributed than the IS2 data, which are acquired with a far from random distribution. Indeed, straight-line arrangements may give rise to biases and inaccuracies in the kriging predictions, as it may lead to overfitting in densely sampled regions. The only way to deal with this, however, would be to entirely remove parts of the IS2 data, which is not a desired solution.

P2l75, Significant disparities exist between LRM data and SARIn data. The authors should expound on their considerations regarding the consistency between these two datasets. Providing insights into the strategies employed to address or account for these differences would contribute to a more thorough understanding of the data integration process and potential impacts on the study's outcomes.

We only include LRM data for a very tiny region. The number of datapoints included consists of 0.1% of the total data set, and does not have any major impact on the study. We decided to include it in order to provide CS2 data for the entire 50km of the margin.

P3l83, A higher sampling density often correlates with improved elevation simulation results. In the context mentioned, it would be beneficial to clarify the purpose of avoiding oversampling and elaborate on the potential consequences associated with it. Understanding the reasoning behind this choice and its potential impacts would enhance transparency regarding the sampling strategy employed in the study.

Ideally, data points should exhibit spatial variability representative of the underlying field, capturing both local fluctuations and broader trends. Straight-line arrangements can introduce biases or inaccuracies into

the kriging estimates, as it may lead to overfitting in densely sampled regions, where local variations are excessively emphasized, while neglecting broader trends. A balanced distribution of data, capturing both local and global variability is therefore essential for robust kriging interpolation.

The straight-line arrangement of the IS2 data cannot be changed. However, to obtain a more balanced distribution we remove data from the weak IS2 beams, which are located right next to the strong beams. Retaining these data would provide no new information, but lead to excessive oversampling.

P3l88, However, such downscaling might diminish the inherent advantages of IS2. Additionally, in the case of CS2 data, it is essential to elucidate how the authors addressed the potential impact of observation resolution. Clarifying the strategies employed to mitigate the effects of observation resolution on CS2 data would contribute to a more comprehensive understanding of the data processing methodology.

Agreed, to some extent the downscaling diminishes the inherent advantages of IS2. It was, however, necessary to do this downscaling in order to assimilate the data set with the CS2 data, see also our answer to one of the previous comments. We do not see any need to mitigate the lower resolution of the CS2 data, as we do not attempt to create the elevation anomaly field in higher resolution that the approximate resolution of the input data.

P3l93, it is evident that IS2 data plays a more central role in DEM generation. Consequently, it raises the question: What specific role does CS2 serve in this context? Considering the non-uniform distribution of CS2, there arises a concern about the potential introduction of uncertainty into elevation estimations. Addressing the distinct contributions and potential uncertainties associated with CS2 data would enhance clarity regarding its significance in the overall DEM generation process.

The CS2 data provides 25% of the total elevation data set, which in our opinion is not a negligible portion.

We have in the new version of the manuscript improved the uncertainty estimation of both data sets.

P4l118, the impact of CS2 signal penetration depth in snow is not explicitly addressed in this section. Providing insights into how the authors considered and accounted for this factor would enhance the completeness of the discussion.

This is not relevant since the PRODEMs are based on data from the ice sheet margin during summer, and therefore largely covers only the snow-free zone.

P4l124, the reason for choosing a 500m resolution is not explicitly stated. Including the rationale behind this resolution choice would provide clarity and context for readers seeking to understand the decision-making process.

This is now more clearly stated:

*The PRODEMs are constructed using Point Kriging, in which approach the interpolated elevation anomaly field is evaluated at the grid cell centres. For this approach to accurately represent the mean field value within a grid cell, the grid resolution must align with the scale of variability of the observation data. After downsampling of the IS2 data, both CS2 and IS2 data sets are representative for an area of a few hundred meters, and an appropriate resolution for the anomaly field of the interpolated PRODEMs is therefore on the order of 500 m.*

P5l146, the uncertainty associated with Kriging interpolation is not specified. Offering details on the uncertainties inherent in the Kriging interpolation method would contribute to a more thorough understanding of the potential limitations in the DEM generation process.

The uncertainty associated with Kriging interpolation is calculated based on the equation provided in L436.

*P7l236, the stated measurement precision applies specifically to the interior areas of Antarctica. It's crucial to note that for steep terrains, the accuracy is expected to be lower. Emphasizing this distinction would provide a more nuanced understanding of the precision associated with different topographical features.*

As correctly pointed out, the stated measurement precision specifically applies to the flat interior areas of Antarctica, and we have added this to the sentence (the addition in red):

*The measurement precision of the IS2 ATL06 product has been documented to be as small as 9cm over the flat interior part of Antarctica.*

Note, however, that we do not use this value in the subsequent analysis, but instead use elevation differences from crossing satellite tracks to provide an estimated uncertainty for the individual observations depending (primarily) on terrain roughness. In areas of low roughness, we estimate an instrument measurement uncertainty of 8cm, which is in accordance with the results from the interior of Antarctica.

*Section 4.1, The calculation method for DEM uncertainty, particularly how the authors integrated data from the two altimeter datasets, is not clearly presented in this section. Including details on the calculation method would enhance transparency and facilitate a more comprehensive assessment of the uncertainty analysis.*

We have revised the uncertainty model for the individual observations, so that the spatial (including geolocation and instrument measurement uncertainty) and temporal components now are separately defined. The total uncertainty on an observation is found by adding the two uncertainty components in quadrature. The text in this section has been updated accordingly.

The DEM uncertainty from the kriging interpolation is calculated based on equation 5 (L436), which allows to account for the different uncertainty of the surrounding observations.

*P8l279, Notably, there appears to be a relatively large elevation difference between CS2 and IS2. It is pertinent to inquire whether the authors have considered any correction methods to address or minimize this discrepancy. A discussion on potential strategies or considerations in correcting the observed elevation differences would contribute to a more thorough evaluation of the data integration process.*

Indeed, in some places we observe quite some elevation differences between closely-spaced (50m) CS2 and IS2 observations. It is worth noting that the outliers in this distribution primarily arise from satellite track crossings in areas of high roughness, where the terrain may vary significantly within a short distance, and a substantial elevation difference therefore is to be expected. Investigating the spread of the distribution as function of terrain roughness (in a similar manner to the analysis performed for estimating the spatial uncertainty of CS2 and IS2), we find a linear increase in dispersion (MAD) of the elevation difference distribution with roughness (see the figure below). At small roughness values (roughness < 2), the dispersion of the distribution is described by a MAD value of 0.48m (average value), corresponding to a $1\sigma$=0.50m. This number should be seen in relation to the spatial uncertainty of the CS2 and IS2 observations at small roughness values, which is 0.39m (CS2) and 0.08 (IS2), respectively. Combining the two in quadrature, we therefore expect an uncertainty of their associated elevation differences of 0.40m. The spread of the elevation difference distribution can therefore largely be attributed the individual uncertainty of the two observations.

In regard to estimating any bias, we must instead consider the mean or median of the distribution. Considering the latter to provide the best estimate for the bias, we notice that its value is lowest in areas of low roughness (-5 cm). This is despite these smooth areas often being located at high-elevation central areas, which may periodically be covered by snow during the designated summer period, which we expect to lead to some, minor, differences in the measured elevation from the two satellites. We therefore consider 5cm to be an upper value for the potential bias.

It is worth noting that this value shows significant spatial variability; magnitude as well as sign differs when conducting the analysis on drainage basin scale: The median value ranges from -0.26 to +0.40 depending on drainage basin, although it should be noted that the analysis for some basins is hampered by poor statistics.

We therefore limit ourselves to note that the maximum potential bias value of 5cm bias is well within the uncertainties of the CS2 datapoints. In accordance with previous literature, we therefore cannot identify any significant bias between the two sets of satellite observations, and consequently we cannot justify to make any bias correction.

We have revised Section 4.2 to include more information on the bias estimation procedure, and a new figure (replacing Fig. 3a).

[Figure]

Figure 4: a) Distribution of elevation differences, computed as $dh = h_{cs2} - h_{is2}$, near intersecting CS2 and IS2 satellite tracks, when imposing strict limits in terms of maximum distance and time between acquisitions. b) Evolution of median and MAD of the resulting elevation difference distribution as function of terrain roughness. Data from the PRODEM area during summers 2019 through 2022.

P9l292, it would be beneficial to expound on the specific steps involved in DETREND. Additionally, clarification on the basis for employing linear interpolation and whether it introduces any uncertainties would be useful.

We decided to do the detrending in the simplest possible way using linear interpolation in the ArcticDEM grid. We note that linear interpolation tends to conserve the general trend of data between grid points, making it suitable for interpolating values within a regular grid, and we have no reason to believe that a more sophisticated method would perform any better.

P9l298, the term "ELEVATION ANOMALIES" requires clarification, and providing details on the proportion of outliers would offer a more precise evaluation of the data characteristics.

We have added to following (in red) to L293:

*Prior to interpolation, the satellite altimetry data are detrended by subtracting a reference DEM (ArcticDEM) from each altimetry point measurement using linear interpolation, and the resulting data set is termed elevation anomalies.*

Details on the proportion of outliers are provided in L300-301.

Section 5.3, Explicitly mentioning the software used for Kriging interpolation and listing relevant parameter settings would provide essential information for readers seeking to replicate or understand the interpolation methodology.

We only used the software GSTools to construct the experimental variograms, and all relevant parameter settings are provided in the manuscript (L344-347). For the revised version, we have replaced this software with Scikit-Stat, which allows us to use the Dowd estimator instead of the Cressie estimator (as suggested by reviewer 1).

Section7.1, I strongly recommend incorporating airborne elevation data into the DEM evaluation process. While cross-validation results are valuable, utilizing airborne data for evaluation would provide an additional and crucial perspective on accuracy. This approach ensures a more comprehensive assessment and strengthens the overall validity of the DEM.

We will investigate whether this can be done based on the very limited airborne elevation data available. See comment above.

---

## Author Response (AR2)

**Replies to comments by reviewer 1**

We thank the reviewer for their constructive comments, and we have addressed the points raised as described in following. The reviewer comments are shown in blue, and our point-to-point replies are shown in black.

This is my second review of this paper. Compared to the original version, the authors have made significant improvements, particularly in terms of data sources and methods. Most of the issues I previously raised have been addressed satisfactorily. I recommend that this paper be accepted after minor revisions.

Here are my several comments:

1. The authors emphasize that only in summer can the two types of altimeters obtain consistent elevation observations at the edge of the ice sheet, and they also generated the DEM based on this assumption. While this approach is feasible, why do we need to do this? What is the practical use of obtaining a DEM for the edge of the ice sheet in summer?

While it would indeed be valuable to have annual DEMs for the entire ice sheet, an annually-updated marginal DEM has several use cases. First of all, the margins experience the largest changes, and hence these areas are most important to monitor on short timescales. One specific use case for the PRODEMs is to improve the ice sheet mass balance estimates using the mass budget method, for which the ice sheet mass loss is calculated from the mass flux through gates located across all outlet glaciers. This requires knowledge of ice sheet thicknesses across the fluxgates. Accounting for the annual changes in surface elevation will therefore improve the mass balance estimates, and, not least, its interannual variability. For this purpose, summer DEMs have the added advantage of lesser complexity during density conversions, since we assume that there is no snow or firn.

We have expanded the mentioning of this in the paper to make this use case apparent to the reader. It now reads (additions/changes in red):

*L. 73: The PRODEMs will be* valuable *for marginal mass balance* assessments, *as they provide crucial information on the inter-annual variability in surface elevation across these pivotal areas for understanding the complex interplay between ice sheet, oceanic processes and climate dynamics. Further, one specific use case of the PRODEM series is to improve mass balance assessments for the entire ice sheet based on the mass budget method* (Mankoff et al., 2021; Mouginot et al., 2019; Van Den Broeke et al., 2009)*, in which the solid ice discharge is calculated by summing the contribution from all individual outlet glaciers based on measured ice velocity and ice thickness* (Mankoff et al., 2020). *The annually-changing PRODEM surface elevations will support the ongoing assessment of solid ice discharge, as carried out e.g. within the Programme for Monitoring of the Greenland Ice Sheet (PROMICE) project* (Ahlstrøm & the PROMICE project team, 2008).

2. The authors have analyzed the elevation differences obtained by the two types of altimeters. Although the authors believe that the elevation differences between the two types of altimeters can be ignored, I still see a meter-level deviation between them in Figure 4. Can this deviation be considered negligible? The author needs to further clarify this point or find a way to achieve some consistency between the two datasets. The differences are greater on rougher surfaces, and the edge of the ice sheet is also rough.

In the roughest parts of the PRODEM area, it is not possible to obtain very good statistics when comparing the elevations obtained from the two altimeters near intersecting tracks: Indeed, the

median of the distribution of elevation differences is ~-1 m, but the spread of the distribution is substantial (MAD = 4m), indicating that the median difference is not well determined. To quantify this point, we have computed bootstrap confidence intervals for the median of the distribution of elevation differences, and these are added to Figure 4 (see updated figure below).

[Figure]

We note that the median of the dh distribution with 95% confidence is less than 0 m only for areas of extreme roughness. Within these areas, the spatial uncertainty associated with especially the CS2 elevations is large ($\sigma_{cs2}^{spatial} \approx 1.5m$, $\sigma_{is2}^{spatial} \approx 0.3m$; computed based on the equations in Fig. 2b,d). The detected bias is therefore within the margin of the measurement uncertainty, supporting our claim that, even within these areas, the observed bias is not practically significant.

From a physical view, we would also expect the elevation differences to be largest in areas of potential snow cover, e.g. at high altitudes, where the surface tends to be relatively flat. Within these areas, we observe very small elevation differences (5 cm), leading to our statement (L 337) that we consider this value to be an upper limit for the potential bias between the two altimeters over bare ice.

The text has been updated as follows:

*L 330: The median of the distribution tends to be smallest in magnitude in areas of low roughness (-0.05 m; Fig. 4a), where the elevation differences are best determined and the dispersion of the elevation difference distribution is small (Fig. 4b). Only in areas of extreme roughness does the median elevation difference fall outside the 95% confidence interval. This deviation from zero is only a fraction of the large (meter scale) observation uncertainties within these areas (Fig. 2b,d), suggesting that this apparent bias may not reflect a true, systematic difference between the two elevation estimates.*

3. The authors mention in the paper that Kriging interpolation was not used to interpolate the surface elevations but rather the difference between the altimeter elevations and ArcticDEM. What is the physical basis for this?

We consider this approach to have several advantages, as also detailed in the paper. To sum up the two most important points:

- Removing a background field is a standard kriging preprocessing step. Kriging requires the field to be stationary, meaning that it cannot be applied to directly to the elevation field, which displays a clear trend (low elevations in the margin; high elevations in the central

area). By subtracting ArcticDEM we obtain a first-order stationary field, to which we may robustly apply the kriging equations (L 395). We now also mention this earlier in the text where the elevation anomalies are introduced:

*L358: The resulting data set is labelled elevation anomalies. This is a standard preprocessing step prior to kriging (e.g., Dodd et al, 2015, Loonat et al. 2020) to ensure statistical robustness, as the method assumes stationarity of the mean field (see also Section 5.3)*

- The approach allows us to take advantage of the high spatial resolution of ArcticDEM, and to also produce reasonable elevations in areas with limited data (Section 9.1).

4. The author only provides a general explanation for choosing the 500 m resolution. What is the specific reason for choosing this resolution? Has there been a comparison of different resolutions to select the optimal one?

We do not have much choice regarding which resolution to produce the map of elevation anomalies, as we cannot go beyond or below the resolution of the input data: For the applied approach ("Point Kriging"), we must align the grid resolution with the scale of variability of the observations (L197). The resolution of the CS2 data, which is a few hundred meters, is therefore the limiting factor, leading to an appropriate resolution for the elevation anomaly field to be on the order of 500 m (L198).

**Reply to reviewer comments by Romain Hugonnet**

We thank Romain Hugonnet for their constructive comments, and we have addressed the points raised as described in following. The reviewer comments are shown in blue, and our point-to-point replies are shown in black.

**General comments**

The authors have satisfactorily addressed all of my comments, even exploring potential improvements that I had not envisioned, selecting the most performant ones and revising their manuscript accordingly.

In my opinion, the manuscript is almost ready for publication. I just have a minor comment on further refining the text.

**Minor comment**

A lot of text has been added (in the Introduction, Section 4 and Section 7, in particular), with statements that are slightly less polished than the rest of the manuscript. I picked up a few errors in my line-by-line comments below, but probably missed things and didn't want to propose rewordings. Be sure to revise this text thoroughly.

We have thoroughly gone through the text (with primary focus on the introduction, section 4, section 7) and revised the language and structure accordingly.

Additionally, I was not convinced by the large expansion of the introduction (though it followed a comment by referee 2), especially moving the datasets descriptions there. I found it a bit too long to grasp the topic rapidly and getting into the paper. I think it would benefit from being more concise and having the long dataset descriptions separate as originially.

Upon re-reading the current version of the manuscript, we agree with the reviewer's comment – the introduction has become too long. Yet, we also consider it a valid point previously raised by referee 2 that the introduction of the paper would benefit from setting the PRODEMs in context of previously constructed DEMs.

We have therefore revised the introduction as follows: we have removed most of the "brief history of DEMs", and now instead describe the existing range of DEMs in more general terms, with particular focus on how they differ in terms of temporal span and spatial resolution and coverage. The data set descriptions have been moved back to their original place (Section 2).

We hope that both reviewers will be satisfied with this change.

Several sections (for instance the one introducing kriging) are also a bit "educational", and could be cut short when the statistical information is available online (equations or definitions) and the general concept is already introduced.

These sections were expanded in the previous round of reviews to provide further clarity on kriging, as additional explanation was incorporated following feedback from reviewer 1. We consider it very important that the reader is aware of what kriging is, including the prerequisites of using the method, as it is the foundation of the DEM generation. We have therefore decided to keep this section as-is. Regarding the kriging equations: Several versions of the ordinary kriging equations exists in the literature, and we wish to specify which ones we have used.

Finally, and critically, the authors lack a discussion about the application of the methodologies that are at the core of this study: 1/ their data fusion scheme (regionally-varying kriging) and 2/ their uncertainty analysis (estimation of spatially variable errors, correlations). How does those compare to existing studies on the topic applying similar methods to DEMs, dh, or elevation anomalies? Right now, as a non-expert reader (user of PRODEM): either everything seems brand-new, or it is under-referenced. It would largely benefit the study to compare to the existing (=detail where it compares or builds upon existing work, and where it is applying new methods/considerations for this type of application).

We have added the following text to the manuscript at appropriate places to ensure that the reader is aware that some of these methods (or relatively similar ones) have indeed been used in the literature previously, although in a different context.

*L. 204: …if uncertainties are underestimated, the interpolation may overfit the observations while inducing noise in the interpolated field. Nevertheless, the observation uncertainties have been largely disregarded during the construction of existing altimeter-derived ice sheet DEMs.*

*L. 228: To estimate the CS2 spatial uncertainty, we analyse the measured elevation differences at temporally-close intersecting CS2 satellite tracks within our study area, in an approach that lends from the cross-over methods used for deriving local longer-term elevation change rates (Khvorostovsky et al., 2003; Sørensen et al., 2018)*

*L. 357: Prior to interpolation, the satellite altimetry is detrended by subtracting a reference DEM (ArcticDEM v4.1) from each elevation measurement using linear interpolation. The resulting data set is labelled elevation anomalies. This is a standard preprocessing step prior to kriging (e.g., Dodd et al., 2015; Loonat et al., 2020) to ensure statistical robustness, as the method assumes stationarity of the mean field (see also Section 5.3). When using other methods to produce DEMs (e.g., Fan 22), such processing step may not be a requirement.*

We do not know of any other studies applying regionally-varying kriging similar to ours, but we consider it likely to have been used in other fields – although probably in a slightly different implementation, and possibly under a different name. We have therefore added the following sentence to the paper:

*L. 408: For a given location in the PRODEM grid, an appropriate set of variogram parameters is subsequently extracted and applied in the kriging routine (section 5.3.2) to produce an interpolated value for the elevation anomaly. We call this method "regionally-varying kriging".*

*L 883: The PRODEMs are interpolated using regionally varying kriging on elevation anomalies relative to ArcticDEM, in an approach that may also be applicable for interpolation of other semi-stationary fields.*

**Short reply to author response**

On my comment about "dropping the nugget": The authors are right, in their case the nugget also represents the uncertainty, I went a bit too fast here. In the Gaussian Process side of kriging I am also familiar with (which I recommend the authors to dive into if they want to scale their analysis to more dimensions/parametrizations), short-scale variance and uncertainties are always separate parameters and not contained into the same one.

**Line-by-line comments**

232: Dartnell et al. (2000) or Wilson et al. (2007) are the reference for this calculation of the surface roughness

Wilson et al (2007) has been added as reference.

585: Reword colloquial term "cleaned"

Reworded to "filtered"

666: Remove "Click or tap here to enter text"

Removed.

Table 1: Add unit (or specify if it is normalized)

Unit has been added

923-924: "That errors in elevation estimate substantially larger than the prediction uncertainty are much more frequent than theoretically expected" → "elevation estimates"? And yes, but this is only true for a small percentage of the data, which is worth pointing out. Overall this whole sentence could be revised, it's a bit convoluted.

The wording has been changed to the following:

*L 640: The heavier tails, on the other hand, indicate that the reported uncertainties do not capture the full range of variability, and* that, for a small percentage of the data, uncertainties are severely underestimated.

946: "robust" → you mean "consistent"?

Yes, the wording has been changed

Style:

- Quite surprised by first-letter upper-casing, such as for "Kriging" of "Block-LOOCV": I don't think it requires any.

Casing has been changed.

- Defining an acronym also does not require upper-casing either, for example "We use leave-one-out cross-validation (LOOCV)" is correct, and can be continued along the text.

This has been changed

**References**

Wilson, M. F. J., O'Connell, B., Brown, C., Guinan, J. C., & Grehan, A. J. (2007). Multiscale Terrain Analysis of Multibeam Bathymetry Data for Habitat Mapping on the Continental Slope. Marine Geodesy, 30(1–2), 3–35.

Dartnell, P. 2000. Applying Remote Sensing Techniques to map Seafloor Geology/Habitat Relationships. Masters Thesis, San Francisco State University, pp. 108.